# Learning Multi-Agent Communication from Graph Modeling Perspective

**Shengchao Hu[1,2], Li Shen[3]\*, Ya Zhang[1,2], Dacheng Tao[4]**
[1] Shanghai Jiao Tong University, [2] Shanghai AI Laboratory
[3] JD Explore Academy, [4] Nanyang Technological University
{charles-hu,ya_zhang}@sjtu.edu.cn;   {mathshenli,dacheng.tao}@gmail.com

## Abstract

In numerous artificial intelligence applications, the collaborative efforts of multiple intelligent agents are imperative for the successful attainment of target objectives. To enhance coordination among these agents, a distributed communication framework is often employed. However, information sharing among all agents proves to be resource-intensive, while the adoption of a manually pre-defined communication architecture imposes limitations on inter-agent communication, thereby constraining the potential for collaborative efforts. In this study, we introduce a novel approach wherein we conceptualize the communication architecture among agents as a learnable graph. We formulate this problem as the task of determining the communication graph while enabling the architecture parameters to update normally, thus necessitating a bi-level optimization process. Utilizing continuous relaxation of the graph representation and incorporating attention units, our proposed approach, CommFormer, efficiently optimizes the communication graph and concurrently refines architectural parameters through gradient descent in an end-to-end manner. Extensive experiments on a variety of cooperative tasks substantiate the robustness of our model across diverse cooperative scenarios, where agents are able to develop more coordinated and sophisticated strategies regardless of changes in the number of agents.

## 1 Introduction

Multi-agent reinforcement learning (MARL) algorithms play an essential role in solving complex decision-making tasks through the analysis of interaction data between computerized agents and simulated or physical environments. This paradigm finds prevalent application across domains, including autonomous driving (Zhou et al., 2020; Hu et al., 2022), order dispatching (Li et al., 2019; Yang et al., 2018), and gaming AI systems (Peng et al., 2017; Zhou et al., 2023). In the MARL scenarios typically explored in these studies, multiple agents engage in iterative interactions within a shared environment, continually refining their policies through learning from observations to collectively attain a common objective. This problem can be conceptually simplified as an instance of independent RL, wherein each agent regards other agents as elements of its environment. However, the strategies employed by other agents exhibit dynamic uncertainty and evolve throughout the training process, rendering the environment intrinsically unstable from the viewpoint of each individual agent. Consequently, effective collaboration among agents becomes a formidable challenge. Additionally, it's important to note that policies acquired through independent RL are susceptible to overfitting with respect to the policies of other agents, as evidenced by Lanctot et al. (2017).

Communication is a fundamental pillar in addressing this challenge, serving as a cornerstone of intelligence by enabling agents to operate cohesively as a collective entity rather than disparate individuals. Its significance becomes especially apparent when tackling complex real-world tasks where individual agents possess limited capabilities and restricted visibility of the environment (Lajoie et al., 2021; Yu et al., 2022b; Liu et al., 2021). In this work, we consider MARL scenarios wherein the task at hand is of a cooperative nature and agents are situated in a partially observable environment, but each is endowed with different observation power. Each agent is underpinned by a

---

\*Corresponding author: Li Shen

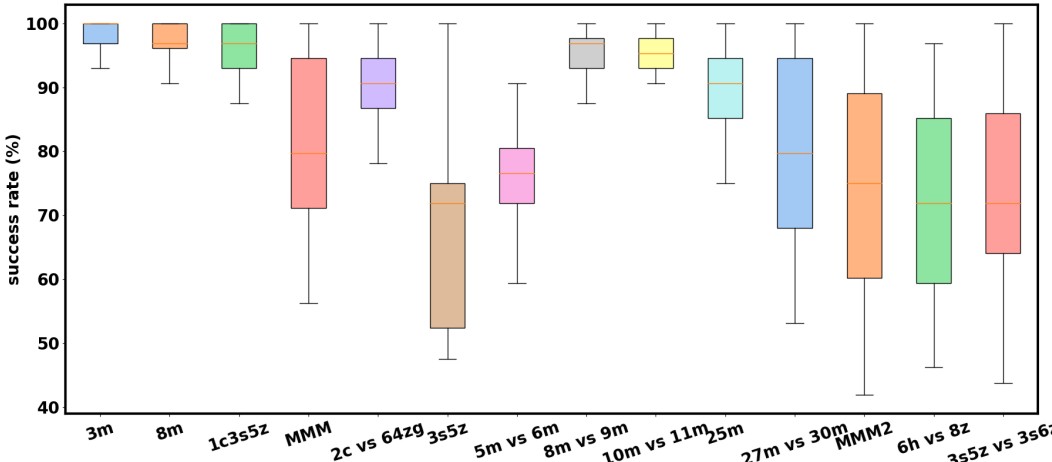

Figure 1: The performance of pre-defined communication architectures across various StarCraftII combat games, each with 10 different seeds. The notable variance observed underscores the importance of searching for the optimal communication architecture.

deep feed-forward network, augmented with access to a communication channel conveying continuous vectors. Considering bandwidth-related constraints, particularly in instances involving wireless communication channels, a limited subset of agents is permitted to exchange messages during each time step to ensure reliable message transfer (Kim et al., 2019). This necessitates meticulous consideration by agents in selecting both the information they convey and the recipient agent.

To facilitate coordinated message exchange, we adopt the centralized training and distributed execution paradigm, as popularized in recent works such as Foerster et al. (2018); Kuba et al. (2022); Yu et al. (2022a), which allows agents access to global information and knowledge of opponents' actions during the training phase. There are several approaches for learning communication in MARL including CommNet (Sukhbaatar et al., 2016), TarMAC (Das et al., 2019), and ToM2C (Wang et al., 2021). However, methods relying on information sharing among all agents or relying on manually pre-defined communication architectures can be problematic. When dealing with a large number of agents, distinguishing valuable information for cooperative decision-making from globally shared data becomes problematic. In such cases, communication may provide limited benefit and could potentially hinder cooperative learning (Jiang & Lu, 2018). Furthermore, in real-world applications, full-scale communication between all agents can be costly, demanding high bandwidth, incurring delays, and imposing significant computational complexity. Manual pre-defined architectures exhibit substantial variance, as evident in Figure 1, which underscores the necessity for meticulous architectural design to achieve optimal communication, as randomly designed architectures may inadvertently hinder cooperation and result in poor overall performance. Dynamic adjustments to the communication graph during inference have garnered significant attention in recent research (Jiang & Lu, 2018; Kim et al., 2019; Wang et al., 2021). However, this approach assumes all agents always need to communicate with one of the other agents, necessitating complex scheduling algorithms, which results in the waste of bandwidth consumption and falls outside the scope of this article.

To address these challenges, we present a novel approach, named CommFormer, designed to facilitate effective and efficient communication among agents in large-scale MARL within partially observable distributed environments. We conceptualize the communication structure among agents as a learnable graph and formulate this problem as the task of determining the communication graph while enabling the architecture parameters to update normally, thus necessitating a bi-level optimization process. In contrast to conventional methods that involve searching through a discrete set of candidate communication architectures, we relax the search space into a continuous domain, enabling architecture optimization via gradient descent in an end-to-end manner. Diverging from previous approaches that often employ arithmetic or weighted means of internal states before message transmission (Peng et al., 2017; Wang et al., 2021), which may compromise communication effectiveness, our method directly transmits each agent's local observations and actions to specific agents based on the learned communication architecture. Subsequently, each agent employs an attention unit to dynamically allocate credit to received messages from the graph modeling perspective, which

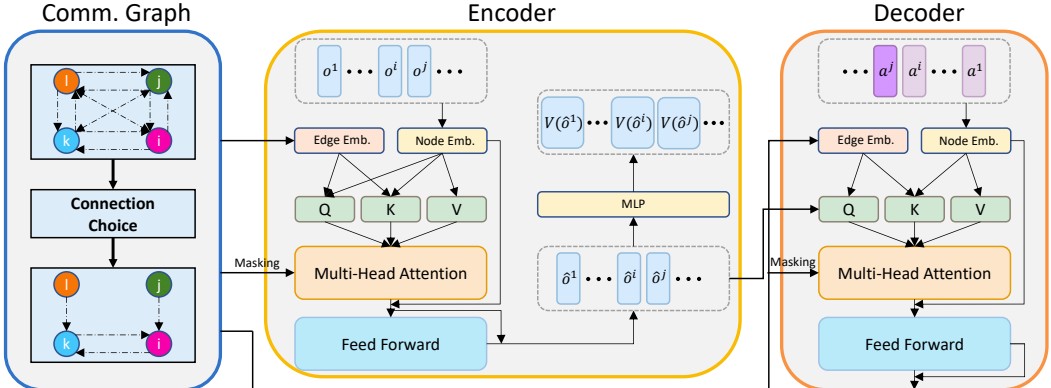

Figure 2: The overview of our proposed CommFormer. CommFormer initiates by establishing the communication graph, which subsequently serves as both the masking and edge embeddings in the encoder and decoder to ensure that agents can exclusively access messages from communicated agents. Subsequently, the encoder and decoder modules come into play, processing a sequence of agents' observations and transforming them into a sequence of optimal actions.

enjoys a monotonic performance improvement guarantee (Wen et al., 2022). Extensive experiments conducted in a variety of cooperative tasks substantiate the robustness of our model across diverse cooperative scenarios. CommFormer consistently outperforms strong baselines and achieves comparable performance to methods allowing information sharing among all agents, demonstrating its effectiveness regardless of variations in the number of agents.

Our contributions can be summarized as follows:

- We conceptualize the communication structure as a graph and introduce an innovative algorithm for learning it through bi-level optimization, which efficiently enables the simultaneous optimization of the communication graph and architectural parameters.

- We propose the adoption of the attention unit within the framework of graph modeling to dynamically allocate credit to received messages, thereby enjoying a monotonic performance improvement guarantee while also improving communication efficiency.

- Through extensive experiments on a variety of cooperative tasks, CommFormer consistently outperforms robust baseline methods and achieves performance levels comparable to approaches that permit unrestricted information sharing among all agents.

## 2 RELATED WORK

**Multi-agent Cooperation**. As a natural extension of single-agent RL, MARL has garnered considerable attention for addressing complex problems within the framework of Markov Games (Yang & Wang, 2020). Numerous MARL methodologies have been developed to tackle cooperative tasks in an online setting, where all participating agents collaborate toward a shared reward objective. To address the challenge of non-stationarity in MARL, algorithms typically operate within two overarching frameworks: centralized and decentralized learning. Centralized methods (Claus & Boutilier, 1998) involve the direct learning of a single policy responsible for generating joint actions for all agents. On the other hand, decentralized learning (Littman, 1994) entails each agent independently optimizing its own reward function. While these methods can handle general-sum games, they may encounter instability issues, even in relatively simple matrix games (Foerster et al., 2017). Centralized training and decentralized execution (CTDE) algorithms represent a middle ground between these two frameworks. One category of CTDE algorithms is value-decomposition (VD) methods, wherein the joint Q-function is formulated as a function dependent on the individual agents' local Q-functions (Rashid et al., 2020; Son et al., 2019; Sunehag et al., 2017). The others (Lowe et al., 2017; Foerster et al., 2018) employ actor-critic architectures and learn a centralized critic that takes global information into account. In this work, we introduce an innovative approach operating under

the CTDE paradigm, with limited communication capabilities, which achieves comparable or even superior performance when compared to these established baselines.

**Communication Learning**. Learning to facilitate communication is a viable approach to enhance multi-agent cooperation. DIAL (Foerster et al., 2016) is the pioneer in proposing learnable communication through back-propagation with deep Q-networks. In this method, each agent generates a message at each timestep, which serves as input for other agents in the subsequent timestep. Building upon this work, a variety of approaches have emerged in the field of multi-agent communication. Some methods adopt pre-defined communication architectures, e.g. CommNet (Sukhbaatar et al., 2016), BiCNet (Peng et al., 2017), and GA-Comm (Liu et al., 2020). These techniques establish fixed communication structures to facilitate information exchange among agents, often utilizing GNN models (Niu et al., 2021; Bettini et al., 2023). In contrast, other approaches such as ATOC (Jiang & Lu, 2018), TarMAC (Das et al., 2019), and ToM2C (Wang et al., 2021) explore dynamic adaptation of communication structures during inference, enabling agents to selectively transmit or receive information. In our research, we align with the former approach, establishing a fixed communication architecture pre-inference. Each agent transmits its local observation and action as a message to a shared channel. Our novel CommFormer approach extends this concept by learning an optimal communication architecture through back-propagation. In contrast to CDC (Pesce & Montana, 2023), which dynamically alters the communication graph through a diffusion process perspective, and TWG-Q (Liu et al., 2022), which emphasizes temporal weight learning and the application of weighted GCN, CommFormer adopts a different approach. It focuses on learning a static graph, aimed at optimizing communication efficiency prior to the inference phase, setting it apart from the traditional methodologies employed by the aforementioned approaches.

## 3 METHOD

The goal of our proposed method is to address the multi-agent collaborative communication problem, which enables agents to operate cohesively as a collective entity rather than disparate individuals. In this paper, we are specifically interested in *learning to construct the communication graph* and *learning how to cooperate with received messages* in a bandwidth-limited way.

### 3.1 PROBLEM FORMULATION

The MARL problems can be modeled by Markov games $\langle \mathcal{N}, \mathcal{O}, \mathcal{A}, R, P, \gamma \rangle$ (Littman, 1994). The set of agents is denoted as $\mathcal{N} = \{1, \ldots, N\}$. The product of the local observation spaces of the agents forms the joint observation space, denoted as $\mathcal{O} = \prod_{i=1}^{n} \mathcal{O}^i$. Similarly, the product of the agents' action spaces constitutes the joint action space, represented as $\mathcal{A} = \prod_{i=1}^{n} \mathcal{A}^i$. The joint reward function, $R : \mathcal{O} \times \mathcal{A} \to [-R_{\max}, R_{\max}]$, maps the joint observation and action spaces to the reward range $[-R_{\max}, R_{\max}]$. The transition probability function, $P : \mathcal{O} \times \mathcal{A} \times \mathcal{O} \to \mathbb{R}$, defines the probability distribution of transitioning from one joint observation and action to another. Lastly, the discount factor, denoted as $\gamma \in [0, 1)$, plays a crucial role in discounting future rewards.

At time step $t \in \mathbb{N}$, an agent $i \in \mathcal{N}$ receives an observation denoted as $o_t^i \in \mathcal{O}^i$. The collection of these individual observations $\boldsymbol{o} = (o^1, \ldots, o^n)$ forms the "joint" observation. Agent $i$ then selects an action $a_t^i$ based on its policy $\pi^i$. It's worth noting that $\pi^i$ represents the policy of the $i^{\text{th}}$ agent, which is a component of the agents' joint policy denoted as $\pi$. Apart from its own local observation $o_t^i$, each agent possesses the capability to receive observations $o_t^j$ from other agents, along with their actions (auto-regressively) $a_t^j$ through a communication channel. At the end of each time step, the entire team collectively receives a joint reward denoted as $R(\mathbf{o}_t, \mathbf{a}_t)$ and observes $\mathbf{o}_{t+1}$, following a probability distribution $P(\cdot|\mathbf{o}_t, \mathbf{a}_t)$. Over an infinite sequence of such steps, the agents accumulate a discounted cumulative return denoted as $R^\gamma \triangleq \sum_{t=0}^{\infty} \gamma^t R(\mathbf{o}_t, \mathbf{a}_t)$.

In practical scenarios where agents have the capability to communicate with each other over a shared medium, two critical constraints are imposed: bandwidth and contention for medium access (Kim et al., 2019). The bandwidth constraint implies that there is a limited capacity for transmitting bits per unit time, and the contention constraint necessitates the avoidance of collisions among multiple transmissions, which is a natural aspect of signal broadcasting in wireless communication. Consequently, each agent can only transmit their message to a restricted number of other agents during each time step to ensure reliable message transfer. In this paper, we conceptualize the communica-

tion architecture as a directed graph, denoted as $\mathcal{G} = \langle \mathcal{V}, \mathcal{E} \rangle$, where each node $v_i \in \mathcal{V}$ represents an agent, and an edge $e_{i \to j} \in \mathcal{E}$ signifies message passing from agent $v_i$ to agent $v_j$. The restriction on communication can be mathematically expressed as the sparsity $\mathcal{S}$ of the adjacency matrix of the edge connections $\alpha$. This sparsity parameter, $\mathcal{S}$, controls the allowed number of connected edges, which is given by $\mathcal{S} \times N^2$, where $N$ is the number of agents.

## 3.2 ARCHITECTURE

The overall architecture of our proposed CommFormer is illustrated in Figure 2.

**Communication Graph.** To design a communication-efficient MARL paradigm, we introduce the Communication Transformer or CommFormer, which adopts a graph modeling paradigm, inspired by developments in sequence modeling (Hu et al., 2023; Wen et al., 2022) We apply the Transformer architecture which facilitates the mapping between the input, consisting of agents' observation sequences $(o^1, \ldots, o^n)$, and the output, which comprises agents' action sequences $(a^1, \ldots, a^n)$. Considering communication constraints, each agent has a limited capacity to communicate with a subset of other agents, represented by the sparsity $\mathcal{S}$ of the adjacency matrix of the edge connections. To identify the optimal communication graph, we treat multiple agents as nodes in a graph and introduce a learnable adjacency matrix, represented by the parameter matrix $\alpha \in \mathbb{R}^{\tilde{N} \times N}$, which are optimized during training in an end-to-end manner.

**Encoder.** The encoder, whose parameters are denoted by $\phi$, takes a sequence of observations $(o^1, \ldots, o^n)$ as input and passes them through several computational blocks. Each such block consists of a *relation-enhanced* mechanism (Hu et al., 2023; Cai & Lam, 2020) and a *multi-layer perceptron* (MLP), as well as *residual connections* to prevent gradient vanishing and network degradation with the increase of depth. In the vanilla multi-head attention, the attention score between the element $o^i$ and $o^j$ can be formulated as the dot-product between their query vector and key vector:

$$s_{ij} = f(o^i, o^j) = o^i W_q^T W_k o^j. \tag{1}$$

$s_{ij}$ can be regarded as implicit information associated with the edge $e_{j \to i}$, where agent $o^i$ queries the information sent from agent $o^j$. To identify the most influential edge contributing to the final performance, we augment the implicit attention score with explicit edge information:

$$\begin{aligned} s_{ij} &= g(o^i, o^j, r_{i \to j}, r_{j \to i}) \\ &= (o^i + r_{i \to j}) W_q^T W_k (o^j + r_{j \to i}), \end{aligned} \tag{2}$$

where $r_{* \to *}$ is obtained from an embedding layer that takes the adjacency matrix $\alpha$ as input. We also apply a mask to the attention scores using the adjacency matrix $\alpha$ to ensure that only information from connected agents is accessible:

$$s_{ij} = \begin{cases} s_{ij}, & e_{j \to i} = 1, \\ -\infty, & e_{j \to i} = 0. \end{cases} \tag{3}$$

We represent the encoded observations as $(\hat{\boldsymbol{o}}^1, \ldots, \hat{\boldsymbol{o}}^n)$, which capture not only the individual agent information but also the higher-level inter-dependencies between agents through communication. To facilitate the learning of expressive representations, during the training phase, we treat the encoder as the critic and introduce an additional projection to estimate the value functions:

$$L_{\text{Encoder}}(\phi) = \frac{1}{Tn} \sum_{m=1}^{n} \sum_{t=0}^{T-1} \left[ R(\mathbf{o}_t, \mathbf{a}_t) + \gamma V_{\bar{\phi}}(\hat{\mathbf{o}}_{t+1}^m) - V_{\phi}(\hat{\mathbf{o}}_t^m) \right]^2, \tag{4}$$

where $\bar{\phi}$ is the target network's parameter, which is a separate neural network that is a copy of the main value function. The update mechanism for $\bar{\phi}$ is executed either through an exponential moving average or via periodic updates in a "hard" manner (Mnih et al., 2015).

**Decoder.** The decoder, characterized by its parameters $\theta$, processes the embedded joint action $\boldsymbol{a}^{0:m-1}, m = 1, \ldots n$ through a series of decoding blocks. The decoding block also incorporates a *relation-enhanced* mechanism for calculating attention between encoded actions and observation representations, along with an MLP and *residual connections*. In addition to the adjacency matrix mask, we apply a constraint that limits attention computation to occur only between agent $i$ and its

preceding agents $j$ where $j < i$. This constraint maintains the sequential update scheme, ensuring that the decoder produces the action sequence in an auto-regressive manner: $\pi_\theta^m(\mathbf{a}^m | \hat{\mathbf{o}}^{1:n}, \mathbf{a}^{1:m-1})$, which guarantees monotonic performance improvement during training (Wen et al., 2022). We apply the PPO algorithm (Schulman et al., 2017) to train the decoder agent:

$$
\begin{aligned}
L_{\text{Decoder}}(\theta) &= -\frac{1}{Tn} \sum_{m=1}^{n} \sum_{t=0}^{T-1} \min\left( \mathbf{r}_t^m(\theta) \hat{A}_t, \text{clip}(\mathbf{r}_t^m(\theta), 1 \pm \epsilon) \hat{A}_t \right), \\
\mathbf{r}_t^m(\theta) &= \frac{\pi_\theta^m(\mathbf{a}_t^m | \hat{\mathbf{o}}_t^{1:n}, \hat{\mathbf{a}}_t^{1:m-1})}{\pi_{\theta_{\text{old}}}^m(\mathbf{a}_t^m | \hat{\mathbf{o}}_t^{1:n}, \hat{\mathbf{a}}_t^{1:m-1})},
\end{aligned}
\tag{5}
$$

where $\hat{A}_t$ is an estimate of the joint advantage function, which can be formulated as $\hat{V}_t = \frac{1}{n} \sum_{m=1}^{n} V(\hat{o}_t^m)$ (Schulman et al., 2015).

### 3.3 TRAINING AND EXECUTION

We employ the CTDE paradigm: during centralized training, there are no restrictions on communication between agents. However, once the learned policies are executed in a decentralized manner, agents can only communicate through a constrained bandwidth channel.

#### 3.3.1 CENTRALIZED TRAINING

During the training stage, we need to determine the communication matrix $\alpha$ while allowing the architecture parameters $\phi$ and $\theta$ to update normally. This implies a bi-level optimization problem (Anandalingam & Friesz, 1992; Colson et al., 2007) with $\alpha$ as the upper level variable and $\phi$ and $\theta$ as the lower-level variable:

$$
\min_{\alpha} \quad \mathcal{L}_{val}(\phi^*(\alpha), \theta^*(\alpha), \alpha),
\tag{6}
$$

$$
\text{s.t.} \quad \phi^*(\alpha), \theta^*(\alpha) = \arg\min_{\phi, \theta} \mathcal{L}_{train}(\phi, \theta, \alpha),
\tag{7}
$$

$$
|\alpha| \leq \mathcal{S} \times N^2,
\tag{8}
$$

where $\mathcal{L} = L_{\text{Encoder}}(\phi) + L_{\text{Decoder}}(\theta)$ with different online rollouts for training $L_{train}$ and validation $L_{val}$, and $|\alpha|$ denotes the number of connected edges. Evaluating the architecture gradient exactly can be prohibitive due to the expensive inner optimization, and each value in $\alpha$ is represented by a discrete value in $\{0, 1\}$. We propose a simple approximation scheme that alternately updates the following formula and relaxes $\alpha$ as a continuous matrix to enable differentiable updating:

$$
\phi = \phi - \xi \nabla_\phi \mathcal{L}_{train}(\phi, \theta, \alpha), \ \theta = \theta - \xi \nabla_\theta \mathcal{L}_{train}(\phi, \theta, \alpha),
\tag{9}
$$

and

$$
\alpha = \alpha - \eta \nabla_\alpha \mathcal{L}_{val}(\phi, \theta, \alpha),
\tag{10}
$$

where $\phi, \theta$ denote the current weights maintained by the algorithm, and $\xi, \eta$ are the learning rate for a step of inner and outer optimization. The idea is to approximate $\phi^*(\alpha), \theta^*(\alpha)$ by adapting $\phi$ and $\theta$ using only a single training step, without fully solving the inner optimization (Equation 7) by training until convergence.

To update the discrete adjacency matrix $\alpha$, we utilize the Gumbel-Max trick (Jang et al., 2016; Maddison et al., 2016) to sample the binary adjacency matrix, which facilitates the continuous representation of $\alpha$ and enables the normal back-propagation of gradients during training. To satisfy constraint 8, we extend the original one-hot Gumbel-Max trick to k-hot, enabling each agent to send messages to a fixed number of $k = \mathcal{S} \times N$ agents:

$$
e_i = \texttt{k\_hot}\big(\text{k-}\arg\max\left[\text{Softmax}(\alpha_{ij} + g_j), \text{ for } j = 1, \ldots, n\right]\big),
\tag{11}
$$

where $g_j$ is sampled from Gumbel(0,1), and $e_i \in \mathbb{N}^N$ represents the edges connected to agent $i$.

#### 3.3.2 DISTRIBUTED EXECUTION

During execution, each agent $i$ has access to its local observations and actions, as well as additional information transmitted by other agents through communication. The adjacency matrix is derived from the parameters $\alpha$ without any randomness as follows:

$$
e_i = \texttt{k\_hot}\big(\text{k-}\arg\max\left[\alpha_{ij}, \text{ for } j = 1, \ldots, n\right]\big).
\tag{12}
$$

Note that each action is generated auto-regressively, in the sense that $a^m$ will be inserted back into the decoder again to generate $a^{m+1}$ (starting with $a^0$ and ending with $a^{n-1}$). Through the use of limited communication, each agent is still able to effectively select actions when compared to fully connected agents, which leads to significant reductions in communication costs and overhead. The overall pseudocode is presented in Algorithm 1.

---

**Algorithm 1** CommFormer

---

1: **Input:** Batch size $B$, number of agents $N$, episodes $K$, steps per episode $T$, sparsity $\mathcal{S}$.
2: **Initialize:** Encoder $\{\phi\}$, Decoder $\{\theta\}$, Replay buffer $\mathcal{B}$, Adjacency matrix $\alpha \in \mathbb{R}^{n \times n}$.
3: **for** $k = 0, 1, \ldots, K - 1$ **do**
4:     **for** $t = 0, 1, \ldots, T - 1$ **do**
5:         Collect a sequence of observations $o_t^1, \ldots, o_t^n$ from environments.
6:         // inference with CommFormer
7:         Generate the matrix $e \in \{0, 1\}^{n \times n}$ according to the $\alpha$ with Equation 12.
8:         Generate representation sequence $\hat{o}_t^1, \ldots, \hat{o}_t^n$ via Encoder $\phi$ with attention score (Equation 2) and mask (Equation 3), similar to the Decoder.
9:         **for** $m = 0, 1, \ldots, n - 1$ **do**
10:            Input $\hat{o}_t^1, \ldots, \hat{o}_t^n$ and $a_t^0, \ldots, a_t^m$ to the Decoder $\theta$ and infer $a_t^{m+1}$ with the auto-regressive manner.
11:        **end for**
12:        Execute joint actions $a_t^0, \ldots, a_t^n$ in environments and collect the reward $R(\boldsymbol{o}_t, \boldsymbol{a}_t)$.
13:        Insert $(\boldsymbol{o}_t, \boldsymbol{a}_t, R(\boldsymbol{o}_t, \boldsymbol{a}_t))$ in to $\mathcal{B}$.
14:     **end for**
15:     // train the CommFormer
16:     Sample a random minibatch of $B$ steps from $\mathcal{B}$.
17:     Generate the matrix $e \in \{0, 1\}^{n \times n}$ according to the $\alpha$ with Equation 11.
18:     Generate $V_\phi(\hat{o})$ with the output layer of the Encoder $\phi$ and compute the joint advantage function $\hat{A}$ based on $V_\phi(\hat{o})$ with GAE.
19:     Input $\hat{o}^1, \ldots, \hat{o}^n$ and $a^0, \ldots, a^{n-1}$, generate $\pi_\theta^1, \ldots, \pi_\theta^n$ at once with the Decoder $\theta$.
20:     Calculate the training loss $L = L_{\text{Encoder}}(\phi) + L_{\text{Decoder}}(\theta)$ with Equation 4 and Equation (5).
21:     Iteratively update the $\phi, \theta$ and $\alpha$ with Equation 9 and Equation 10.
22: **end for**

---

# 4 EXPERIMENT

To evaluate the properties and performance of our proposed CommFormer[1], we conduct a series of experiments using four environments, including Predator-Prey (PP) (Singh et al., 2018), Predator-Capture-Prey (PCP) (Seraj et al., 2022), StarCraftII Multi-Agent Challenge (SMAC) (Samvelyan et al., 2019), and Google Research Football(GRF) (Kurach et al., 2020). A comprehensive description of each environment can be found in the Appendix A. It is worth noting that in certain domains, our objective extends beyond maximizing the average success rate or cumulative rewards. We also aim to minimize the average number of steps required to complete an episode, emphasizing the ability to achieve goals in the shortest possible time.

## 4.1 BASELINES

We compare CommFormer with strong CTDE baselines that do not involve communication, e.g. HAPPO (Yu et al., 2022a), MAPPO (Yu et al., 2022a) and QMIX (Rashid et al., 2020), as well as popular communication methods, e.g. MAGIC (Niu et al., 2021), TarMAC (Das et al., 2019), and QGNN (Kortvelesy & Prorok, 2022) to highlight its effectiveness. Details for each method are provided in Appendix B. During experiments, the implementations of baseline methods are consistent with their official repositories, all hyper-parameters left unchanged at the origin best-performing status. We also include the **fully connected CommFormer** (FC) configuration, where there are no limitations on communication bandwidth. In this configuration, each agent can communicate with all other agents, implying that the sparsity parameter $\mathcal{S}$ is set to 1. FC serves as the upper bound of our methods and demonstrates strong performance on cooperative MARL tasks.

---

[1]Our code is available at: https://github.com/charleshsc/CommFormer

Table 1: Performance evaluations of different metrics and standard deviation on the selected benchmark, where UPDeT's official codebase supports several Marine-based tasks only. Note that the sparsity parameter $\mathcal{S}$ in CommFormer is consistently set to 0.4 for all tasks.

| Task | Difficulty | CommFormer(0.4) | MAT | MAPPO | HAPPO | QMIX | UPDeT | FC | Steps |
|------|-----------|-----------------|-----|-------|-------|------|-------|-----|-------|
| 3m | Easy | **100.0**(0.0) | **100.0**(0.0) | **100.0**(0.4) | **100.0**(1.2) | 96.9(1.3) | **100.0**(5.2) | **100.0**(0.0) | 5e5 |
| 8m | Easy | **100.0**(0.0) | **100.0**(0.0) | 96.8(2.9) | 97.5(1.1) | 97.7(1.9) | 96.3(9.7) | **100.0**(0.0) | 1e6 |
| 1c3s5z | Easy | **100.0**(0.0) | **100.0**(0.0) | **100.0**(2.2) | 97.5(1.8) | 96.9(1.5) | / | **100.0**(0.0) | 2e6 |
| MMM | Easy | **100.0**(0.0) | 83.3(4.8) | 95.6(4.5) | 81.2(22.9) | 91.2(3.2) | / | **100.0**(0.0) | 2e6 |
| 2c vs 64zg | Hard | **100.0**(0.0) | **100.0**(3.1) | **100.0**(2.7) | 90.0(4.8) | 90.3(4.0) | / | **100.0**(3.1) | 5e6 |
| 3s5z | Hard | **100.0**(0.0) | 74.0(6.4) | 72.5(26.5) | 90.0(3.5) | 84.3(5.4) | / | **100.0**(3.1) | 3e6 |
| 5m vs 6m | Hard | 89.6(1.5) | 81.3(5.1) | 88.2(6.2) | 73.8(4.4) | 75.8(3.7) | 90.6(6.1) | **93.8**(4.4) | 1e7 |
| 8m vs 9m | Hard | **100.0**(0.0) | 96.9(0.0) | 93.8(3.5) | 86.2(4.4) | 92.6(4.0) | / | **100.0**(3.1) | 5e6 |
| 10m vs 11m | Hard | **100.0**(1.4) | **100.0**(3.1) | 96.3(5.8) | 77.5(9.7) | 95.8(6.1) | / | **100.0**(0.0) | 5e6 |
| 25m | Hard | **100.0**(0.0) | 0.0(0.1) | **100.0**(2.7) | 0.6(0.8) | 90.2(9.8) | 2.8(3.1) | **100.0**(0.0) | 2e6 |
| 27m vs 30m | Hard+ | 96.9(3.1) | 80.2(3.4) | 93.1(3.2) | 0.0(0.0) | 39.2(8.8) | / | **100.0**(0.0) | 1e7 |
| MMM2 | Hard+ | **100.0**(3.1) | 96.9(0.0) | 81.8(10.1) | 0.3(0.4) | 88.3(2.4) | / | **100.0**(0.0) | 1e7 |
| 6h vs 8z | Hard+ | 96.9(3.1) | 93.8(4.4) | 88.4(5.7) | 0.0(0.0) | 9.7(3.1) | / | **100.0**(0.0) | 1e7 |
| 3s5z vs 3s6z | Hard+ | 87.5(3.1) | 79.2(9.0) | 84.3(19.4) | 82.8(21.2) | 68.8(21.2) | / | **100.0**(3.1) | 2e7 |

| Task | Difficulty | CommFormer(0.4) | QGNN | SMS | TarMAC | NDQ | MAGIC | QMIX | Steps |
|------|-----------|-----------------|------|-----|--------|-----|-------|------|-------|
| 1o2r vs 4r | Hard+ | **96.9**(1.5) | 93.8(2.6) | 76.4 | 39.1 | 77.1 | 22.3 | 51.1 | 2e7 |
| 5z vs 1ul | Hard+ | **100.0**(1.4) | 92.2(1.6) | 59.9 | 44.2 | 48.9 | 0.0 | 82.6 | 1e7 |
| 1o10b vs 1r | Hard+ | 96.9(3.1) | **98.0**(2.9) | 86.0 | 40.1 | 78.1 | 5.8 | 51.4 | 2e7 |

| Task | Metric | CommFormer(0.4) | MAGIC | HetNet | CommNet | I3CNet | TarMAC | GA-Comm | Steps |
|------|--------|-----------------|-------|--------|---------|--------|--------|---------|-------|
| GRF | Success Rate | **100.0**(0.0) | 98.2(1.0) | / | 59.2(13.7) | 70.0(9.8) | 73.5(8.3) | 88.8(3.9) | - |
| | Steps Taken | **25.4**(0.4) | 34.3(1.3) | / | 39.3(2.4) | 40.4(1.2) | 41.5(2.8) | 39.1(3.1) | - |
| PP | Avg. Cumulative $\mathcal{R}$ | **-0.121**(0.008) | -0.386(0.024) | -0.232(0.010) | -0.336(0.012) | -0.342(0.015) | -0.563(0.030) | / | - |
| | Steps Taken | **4.99**(0.31) | 10.6(0.50) | 8.30(0.25) | 8.97(0.46) | 9.69(0.26) | 18.4(0.46) | / | - |
| PCP | Avg. Cumulative $\mathcal{R}$ | **-0.197**(0.019) | -0.394(0.017) | -0.364(0.017) | -0.394(0.019) | -0.411(0.019) | -0.548(0.031) | / | - |
| | Steps Taken | **7.61**(0.66) | 10.8(0.45) | 9.98(0.36) | 11.3(0.34) | 11.5(0.37) | 17.0(0.80) | / | - |

## 4.2 MAIN RESULTS

According to the results presented in Table 1 and Figure 1, our CommFormer with a sparsity parameter $\mathcal{S} = 0.4$ significantly outperforms the state-of-the-art baselines. It consistently finds the optimal communication architecture across diverse cooperative scenarios, regardless of changes in the number of agents. Take the task *3s5z* as an example, where the algorithm needs to control different types of agents: stalkers and zealots. This requires careful design of the communication architecture based on the capabilities of different units. Otherwise, it can even have a detrimental impact on performance, as indicated by the substantial variance displayed in Figure 1. The outcome of *3s5z* presented in Table 1 consistently highlights CommFormer's ability to attain optimal performance with different random seeds, which underscores the robustness and efficiency of our proposed method. Furthermore, in comparison to the FC method, CommFormer nearly matches its performance while retaining only 40% of the edges. This indicates that with an appropriate communication architecture, many communication channels can be eliminated, thereby significantly reducing the hardware communication equipment requirements and expanding its applicability. Finally, it's worth noting that all the results presented in Table 1 are based on the same number of training steps, demonstrating the robustness and effectiveness of our bi-level optimization approach, which consistently converges to the optimal solution while maintaining sample efficiency. The hyper-parameters used in the study and additional detailed results can be found in the Appendix.

## 4.3 ABLATIONS

We conduct several ablation studies, primarily focusing on the SMAC environments, to examine specific aspects of our CommFormer. The parameter $\mathcal{S}$, which determines the sparsity of the communication graph, impacts the number of connected edges. Lower values of $\mathcal{S}$ imply reduced costs associated with communication but may also lead to performance degradation. Additionally, we conduct ablation studies to investigate the essence of architecture searching, where we generate various pre-defined architectures using different random seeds, simulating manually pre-defined settings.

**Sparsity.** The parameter $\mathcal{S}$ introduced in our bi-level optimization controls the number of connected edges, ensuring that it does not exceed $\mathcal{S} \times N^2$, as specified in Equation 8. To simplify this constraint, we ensure that the total number of edges $|\alpha|$ equals $\mathcal{S} \times N^2$, with each agent communicating with a fixed number of $\mathcal{S} \times N$ agents. Smaller values of $\mathcal{S}$ reduce the cost associated with communication but may also result in performance degradation. To investigate the impact of varying $\mathcal{S}$, we conduct

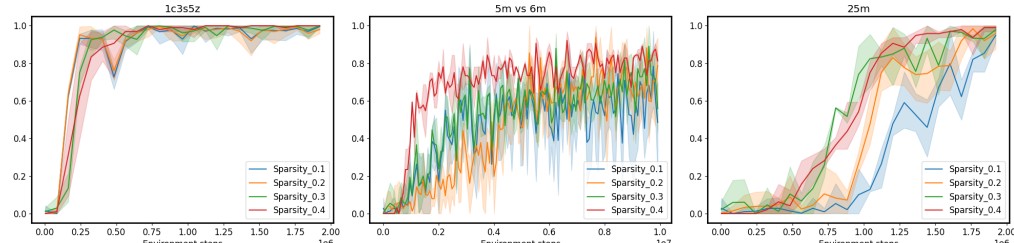

Figure 3: Performance comparison on SMAC tasks with different sparsity $\mathcal{S}$. Note that as the value of sparsity $\mathcal{S}$ gradually increases, the performance of CommFormer improves across various environments. This effect is particularly pronounced in environments with a large number of agents.

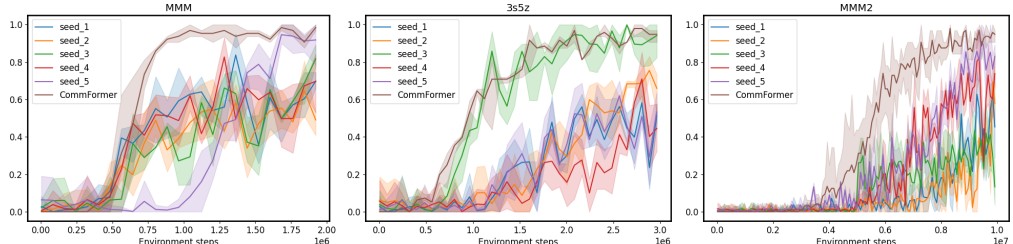

Figure 4: Performance comparison on SMAC tasks with different manually pre-defined communication architectures. CommFormer consistently achieves optimal performance, which underscores its capability to autonomously search for the optimal communication architecture, highlighting its adaptability across various scenarios and tasks.

a series of experiments, whose results are presented in Figure 3. For simpler tasks, such as *1c3s5z*, achieving a 100% win rate is possible even when each agent can only communicate with one other agent. Nevertheless, As task complexity and the number of participating agents increase, a larger value of sparsity $\mathcal{S}$ becomes necessary to attain superior performance.

**Architecture Searching.** In light of the constraints imposed by limited bandwidth and contention for medium access, designing the communication architecture for each agent becomes a critical task. To investigate the impact of different communication graph configurations, we conduct experiments using various random seeds, simulating different individuals' approaches to the problem. The results, as depicted in Figure 1 and 4, highlight that manually pre-defining the communication architecture often leads to significant performance variance, demanding expert knowledge for achieving better results. In contrast, our proposed method leverages the continuous relaxation of the graph representation. This innovative approach allows for the simultaneous optimization of both the communication graph and architectural parameters in an end-to-end fashion, all while maintaining sample efficiency. This underscores the essentiality and effectiveness of our approach in tackling the challenges of multi-agent communication in constrained environments.

## 5 CONCLUSION

In this paper, we introduce a novel approach called CommFormer, which addresses the challenge of learning multi-agent communication from a graph modeling perspective. Our approach treats the communication architecture among agents as a learnable graph and formulates this problem as the task of determining the communication graph while enabling the architecture parameters to update normally, thus necessitating a bi-level optimization process. By leveraging continuous relaxation of graph representation and incorporating attention mechanisms within the graph modeling framework, CommFormer enables the concurrent optimization of the communication graph and architectural parameters in an end-to-end manner. Extensive experiments conducted on a variety of cooperative tasks illustrate the significant performance advantage of our approach compared to other state-of-the-art baseline methods. In fact, CommFormer approaches the upper bound in scenarios where unrestricted information sharing among all agents is permitted. We believe that our work opens up new possibilities for the application of communication learning in the field of MARL, where effective communication plays a pivotal role in addressing various challenges.

ETHICS STATEMENTS

This paper does not raise any ethics concerns. This study does not involve any human subjects, practices to data set releases, potentially harmful insights, methodologies and applications, potential conflicts of interest and sponsorship, discrimination/bias/fairness concerns, privacy and security issues, legal compliance, and research integrity issues.

ACKNOWLEDGMENTS

This work is supported by the National Key R&D Program of China (No. 2022ZD0160702), STCSM (No. 22511106101, No. 22511105700, No. 21DZ1100100), 111 plan (No. BP0719010) and National Natural Science Foundation of China (No. 62306178).

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

# Appendices

## A    DETAILED DESCRIPTION OF ENVIRONMENTS.

During the main experiments, we compare our method within four environments, including Predator-Prey (PP) (Singh et al., 2018), Predator-Capture-Prey (PCP) (Seraj et al., 2022), StarCraftII Multi-Agent Challenge (SMAC) (Samvelyan et al., 2019), and Google Research Football(GRF) (Kurach et al., 2020).

- **PP**. The goal is for $N$ predator agents with limited vision to find a stationary prey and move to its location. The agents in this domain all belong to the same class (i.e., identical state, observation and action spaces).

- **PCP**. We have two classes of predator and capture agents. Agents of the predator class have the goal of finding the prey with limited vision (similar to agents in PP). Agents of the capture class, have the goal of locating the prey and capturing it with an additional capture-prey action in their action-space, while not having any observation inputs (e.g., lack of scanning sensors).

- **SMAC**. In these experiments, CommFormer controls a group of agents tasked with defeating enemy units controlled by the built-in AI. The level of combat difficulty can be adjusted by varying the unit types and the number of units on both sides. We measure the winning rate and compare it with state-of-the-art baseline approaches. Notably, the maps *1o10b_vs_1r* and *1o2r_vs_4r* present formidable challenges attributed to limited observational scope, while the map *5z_vs_1ul* necessitates heightened levels of coordination to attain successful outcomes.

- **GRF**. We evaluate algorithms in the football *academy scenario 3 vs. 2*, where we have 3 attackers vs. 1 defender, and 1 goalie. The three offending agents are controlled by the MARL algorithm, and the two defending agents are controlled by a built-in AI. We find that utilizing a 3 vs. 2 scenario challenges the robustness of MARL algorithms to stochasticity and sparse rewards.

## B    DETAILED BASELINES

We compare our CommFormer with strong baselines without communication, and popular communication methods to showcase the effectiveness. During the SMAC environments, the baseline methods are as follows, each of them is based on the CTDE paradigm to ensure fair comparison: (1) **MAPPO** (Yu et al., 2022a) directly apply PPO in MARL and use one shared set of parameters for all agents, without any communication. (2) **HAPPO** (Kuba et al., 2022) implement multi-agent trust-region learning by the sequential update scheme with a monotonic improvement guarantee. (3) **QMIX** (Rashid et al., 2020) incorporates a centralized value function to facilitate decentralized decision-making and efficient coordination among agents while addressing credit assignment issues. (4) **UPDeT** (Hu et al., 2021) decouples each agent's observations into a sequence of observation entities and uses a Transformer to match different action-observation. (5) **MAT** (Wen et al., 2022) treats cooperative MARL as sequence modeling and adopts a fixed encoder and a fully decentralized actor for each individual agent. (6) **SMS** (Xue et al., 2022) calculates the Shapley Message Value to explicitly evaluate each message's value, learning an efficient communication protocol in more complex scenarios. (7) **TarMAC** (Das et al., 2019) utilizes an attention mechanism to integrate messages according to their relative importance. (8) **NDQ** (Wang et al., 2019) aims at learning nearly decomposable Q functions via communication minimization. (9) **MAGIC** (Niu et al., 2021) makes use of hard attention to construct a dynamic communication graph, which then combines with a graph attention neural network to process the messages. (10) **QGNN** (Kortvelesy & Prorok, 2022) introduces a value factorisation method that uses a graph neural network based model.

For other domains, we benchmark our approach against a variety of state-of-the-art communication-based MARL baselines: (1) **CommNet** (Sukhbaatar et al., 2016) uses continuous communication for fully cooperative tasks, where the model consists of multiple agents and the communication between them is learned alongside their policy. (2) **I3CNet** (Singh et al., 2018) controls continuous communication with a gating mechanism and uses individualized rewards for each agent to gain better performance and scalability while fixing credit assignment issues. (3) **GA-Comm** (Liu et al., 2020) models the relationship between agents by a complete graph and proposes a novel game ab-

straction mechanism based on two-stage attention network. (4) **HetNet** (Seraj et al., 2022) learns efficient and diverse communication models for coordinating cooperative heterogeneous teams based on heterogeneous graph-attention networks.

## C    HYPER-PARAMETER SETTINGS

During our experiments, we maintain consistency in the implementations of baseline methods by using their official repositories, and we keep all hyperparameters unchanged from their original best-performing configurations. Specific hyperparameters used for different algorithms and tasks can be found in Tables 2 to 4. To ensure a fair comparison and validate that CommFormer achieves optimal performance without compromising sample efficiency, we adopted the same hyperparameter settings as MAT (Wen et al., 2022).

## D    DETAILS OF EXPERIMENTAL RESULTS

We provide detailed training figures (Figure 6) for various methods to substantiate our claim that our approach facilitates simultaneous optimization of the communication graph and architectural parameters in an end-to-end manner, all while preserving sample efficiency.

## E    MORE VISUAL RESULTS

We present additional visual results (Figure 5) that showcase the final communication architecture obtained through our search process. These visualizations offer a more intuitive understanding of the architecture's evolution during training. As training progresses, the communication structure adapts to improve performance. Additionally, as we move towards the later stages of training, the model's architecture stabilizes, with only minimal changes observed, typically involving 1 or 2 edges.

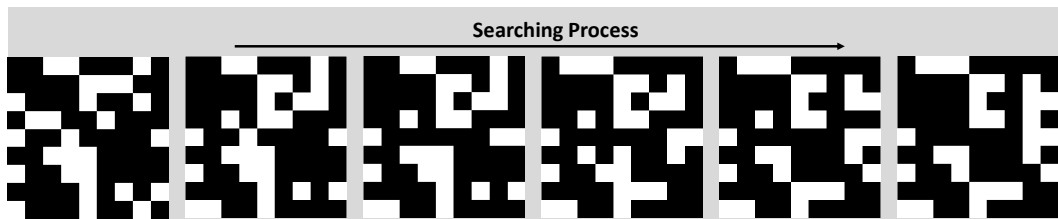

Figure 5: The searching process of CommFormer in the SMAC task *1c3s5z*. In this representation, a white square corresponds to a value of 1, indicating the presence of an edge connection.

## F    APPLICATION CONSIDERATION

A possible application of this study is to create an efficient communication framework tailored for enclosed, finite environments, typical of logistics warehouses. In these settings, agent movement is limited to designated zones, and communication is facilitated through overhead wires, akin to a trolleybus system.

In contrast, open environments present unique challenges, primarily due to the potential vast distances between agents, which require wireless communication and may hinder effective communication. To address this, a straightforward approach could be to add bidirectional edges between agents when they come within close proximity, enabling communication between them (Seraj et al., 2022). However, a more effective solution may involve a hybrid approach that considers the constraint on the available bandwidth: initially segmenting agents into groups based on proximity, followed by an internal search for an optimal communication graph within each group. If agent distances vary dynamically during testing, this process is repeated as necessary to adjust the communication graph in real time, ensuring continuous adaptability to changing environmental conditions.

Table 2: Common hyper-parameters used for our method in the experiments.

| hyper-parameters | value | hyper-parameters | value | hyper-parameters | value |
|---|---|---|---|---|---|
| critic lr | 5e-4 | actor lr | 5e-4 | use gae | True |
| gain | 0.01 | optim eps | 1e-5 | batch size | 3200 |
| training threads | 16 | num mini-batch | 1 | rollout threads | 32 |
| entropy coef | 0.01 | max grad norm | 10 | episode length | 100 |
| optimizer | Adam | hidden layer dim | 64 | use huber loss | True |

Table 3: Specific hyper-parameters used for our method in the experiments.

| hyper-parameters in PP | value | hyper-parameters in PCP | value | hyper-parameters in GRF | value |
|---|---|---|---|---|---|
| Number Agents | 3 | Number Predators | 2 | Number Agents | 3 |
| Number Enemies | 1 | Number Captures | 1 | eval episode length | 200 |
| vision | 1 | Number Enemies | 1 | - | - |
| eval episode length | 20 | vision | 1 | - | - |
| - | - | eval episode length | 20 | - | - |

Table 4: Different hyper-parameters used for CommFormer in different tasks.

| tasks | ppo epochs | ppo clip | num blocks | num heads | stacked frames | steps | $\gamma$ |
|---|---|---|---|---|---|---|---|
| 3m | 15 | 0.2 | 1 | 1 | 1 | 5e5 | 0.99 |
| 8m | 15 | 0.2 | 1 | 1 | 1 | 1e6 | 0.99 |
| 1c3s5z | 10 | 0.2 | 1 | 1 | 1 | 2e6 | 0.99 |
| MMM | 15 | 0.2 | 1 | 1 | 1 | 2e6 | 0.99 |
| 2c vs 64zg | 10 | 0.05 | 1 | 1 | 1 | 5e6 | 0.99 |
| 3s vs 5z | 15 | 0.05 | 1 | 1 | 4 | 5e6 | 0.99 |
| 3s5z | 10 | 0.05 | 1 | 1 | 1 | 3e6 | 0.99 |
| 5m vs 6m | 10 | 0.05 | 1 | 1 | 1 | 1e7 | 0.99 |
| 8m vs 9m | 10 | 0.05 | 1 | 1 | 1 | 5e6 | 0.99 |
| 10m vs 11m | 10 | 0.05 | 1 | 1 | 1 | 5e6 | 0.99 |
| 25m | 15 | 0.05 | 1 | 1 | 1 | 2e6 | 0.99 |
| 27m vs 30m | 5 | 0.2 | 1 | 1 | 1 | 1e7 | 0.99 |
| MMM2 | 10 | 0.05 | 1 | 1 | 1 | 1e7 | 0.99 |
| 6h vs 8z | 15 | 0.05 | 1 | 1 | 1 | 1e7 | 0.99 |
| 3s5z vs 3s6z | 5 | 0.05 | 1 | 1 | 1 | 2e7 | 0.99 |
| 1o10b vs 1r | 10 | 0.2 | 1 | 1 | 1 | 2e7 | 0.99 |
| 1o2r vs 4r | 5 | 0.05 | 1 | 1 | 1 | 1e7 | 0.99 |
| 5z vs 1ul | 10 | 0.05 | 1 | 1 | 1 | 1e7 | 0.99 |
| PP | 10 | 0.05 | 1 | 1 | 1 | 1e7 | 0.99 |
| PCP | 10 | 0.05 | 1 | 1 | 1 | 1e7 | 0.99 |
| GRF | 10 | 0.05 | 1 | 1 | 1 | 1e7 | 0.99 |

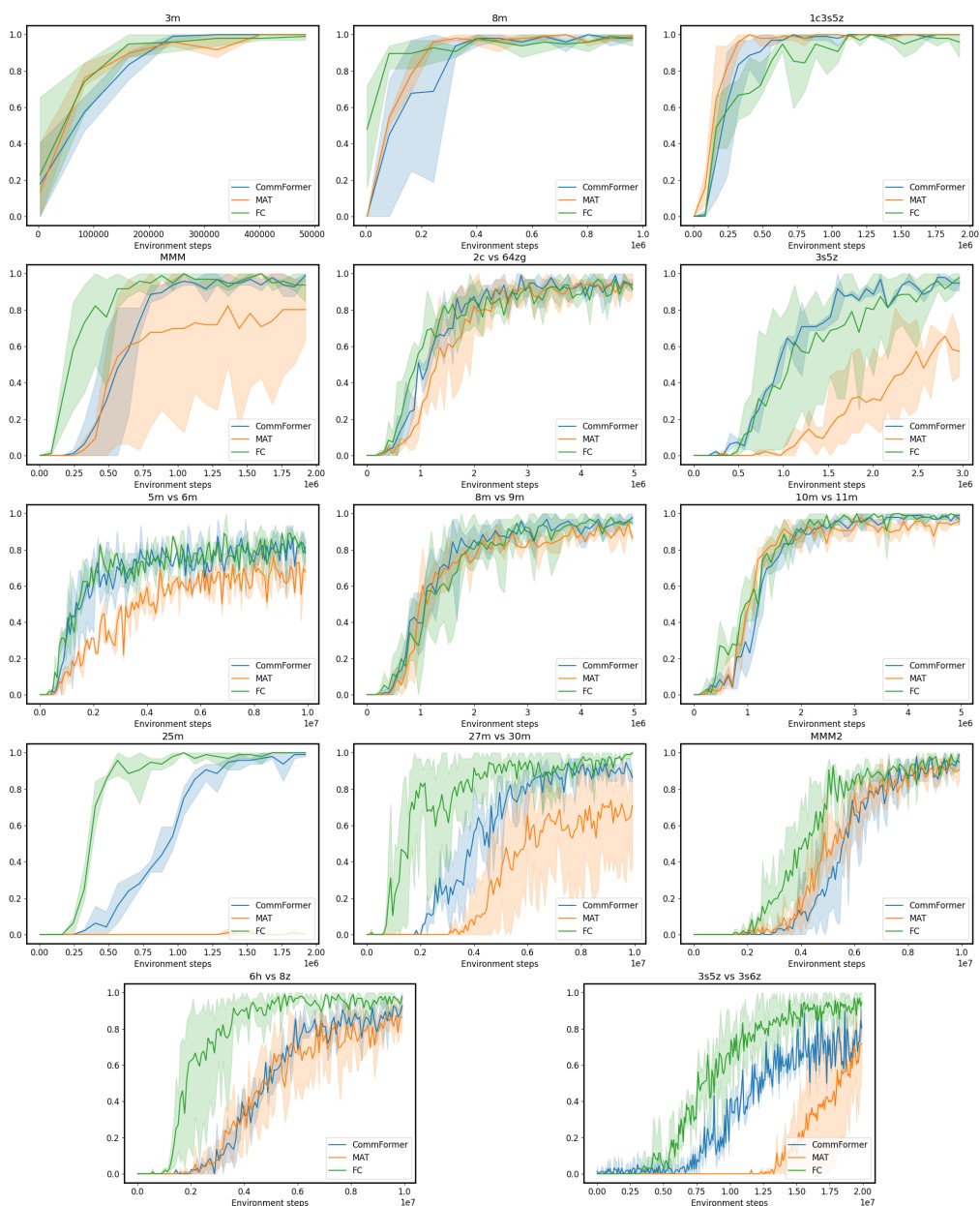

Figure 6: Performance comparison on SMAC tasks. CommFormer consistently outperforms strong baselines and achieves comparable performance to methods allowing information sharing among all agents, demonstrating its effectiveness regardless of variations in the number of agents.

