# OpenReview forum: "Learning Multi-Agent Communication from Graph Modeling Perspective"
_ICLR.cc/2024/Conference — ICLR 2024 poster_

### Official Review · Reviewer_WiJp · 2023-10-31

**Soundness:** 3 good
**Presentation:** 2 fair
**Contribution:** 2 fair
**Rating:** 6
**Confidence:** 3

**Summary:**

This paper introduces a novel approach called CommFormer, which addresses the challenge of learning multi-agent communication from a graph modeling perspective. The communication architecture among agents is modelled as a learnable graph. The problem is treated as the task of determining the communication graph while enabling the architecture parameters to update normally, thus necessitating a bi-level optimization process. By leveraging continuous relaxation of graph representation and incorporating attention mechanisms within the graph modeling framework, CommFormer enables the concurrent optimization of the communication graph and architectural parameters in an end-to-end manner.

**Strengths:**

This paper introduces a novel approach which models the communication architecture among agents as a learnable graph. The considered problem is formulated as the task of determining the communication graph while enabling the architecture parameters to update normally, thus necessitating a bi-level optimization process.

**Weaknesses:**

There have been some works which learns multi-agent cooperative behaviors based on learnable graphs. It would be better to illustrate the differences of the paper compared to them. An example is provided below.

Liu, Y., Dou, Y., Li, Y., Xu, X., & Liu, D. (2022). Temporal Dynamic Weighted Graph Convolution for Multi-agent Reinforcement Learning. Proceedings of the Annual Meeting of the Cognitive Science Society.

**Questions:**

1. The paper proposes a communication-based MARL method. In fact the paradigm CTDE is not suited for such method. There are still some communications among agents for the execution of policies. It seems that CTCE is more suited for the proposed method. Some CTCE based MARL methods. for example, graph-based MARL methods, should be considered for the comparison in the experiment.

2. In Table 1, the FC is a little bit confusing. Even there are no constrictions on the communication bandwidth, the win rate is still hard to be 100% as it depends how the opponents perform. Of course, 100% is the maximum value for the win rate, but it is a meaningless upper bound. Further, how the value 93.8 is obtained in FC column as the upper bound?

---

> ### Author Response · Authors · 2023-11-17
>
> ### Q1
> > Some CTCE based MARL methods. for example, graph-based MARL methods, should be considered for the comparison in the experiment.
>
> Thanks for the suggestion! We have incorporated additional experiments to enhance the generalization of our method. Taking into account the communication domains explored in previous works, we have included the following experiments. It is worth noting that in certain domains, our objective extends beyond maximizing the average success rate or cumulative rewards. We also aim to minimize the average number of steps required to complete an episode, emphasizing the ability to achieve goals in the shortest possible time.
> + Another three maps in the SMAC environment: 1o10b_vs_1r and 1o2r_vs_4r, which pose challenges due to partial observability, and 5z_vs_1ul, where successful outcomes require strong coordination.
> + Predator-Prey (PP) [1]: The goal is for 𝑁 predator agents with limited vision to find a stationary prey and move to its location. The agents in this domain all belong to the same class (i.e., identical state, observation and action spaces).
> + Predator-Capture-Prey (PCP) [2]: We have two classes of predator and capture agents. Agents of the predator class have the goal of finding the prey with limited vision (similar to agents in PP). Agents of the capture class, have the goal of locating the prey and capturing it with an additional capture-prey action in their action-space, while not having any observation inputs (e.g., lack of scanning sensors).
> + Google Research Football (GRF) [11]: We evaluate algorithms in the football academy scenario 3 vs. 2, where we have 3 attackers vs. 1 defender, and 1 goalie. The three offending agents are controlled by the MARL algorithm, and the two defending agents are controlled by a built-in AI. We find that utilizing a 3 vs. 2 scenario challenges the robustness of MARL algorithms to stochasticity and sparse rewards.
>
> We include several state-of-the-art graph-based multi-agent communication learning approaches as additional baselines in our evaluation. These methods encompass QGNN [7], SMS [3], TarMAC [4], NDQ [5], MAGIC [6], HetNet [2], CommNet [8], I3CNet [9], and GA-Comm [10].
>
> The performance results of these baselines are presented below. It is important to note that due to time constraints, we directly obtain the performance results from the respective papers. Our CommFormer consistently demonstrates favorable performance across the evaluated metrics.
>
>
>
> | Task        | Metric       | CommFormer(0.4) | QGNN           | SMS  | TarMAC | NDQ  | MAGIC | QMIX |
> | ----------- | ------------ | --------------- | -------------- | ---- | ------ | ---- | ----- | ---- |
> | 1o2r_vs_4r  | Success Rate | **96.9 $\pm$ 1.5**  | 93.8 $\pm$ 2.6 | 76.4 | 39.1   | 77.1 | 22.3  | 51.1 |
> | 1o10b_vs_1r | Success Rate | 96.9 $\pm$ 3.1  | **98.0 $\pm$ 2.9** | 86.0 | 40.1   | 78.1 | 5.8   | 51.4 |
> | 5z_vs_1ul   | Success Rate | **100.0 $\pm$ 1.4** | 92.2 $\pm$ 1.6 | 59.9 | 44.2   | 48.9 | 0.0   | 82.6 |
>
> | Task | Metric       | CommFormer(0.4) | MAGIC           | CommNet        | I3CNet         | TarMAC         | GA-Comm        |
> | ---- | ------------ | --------------- | --------------- | -------------- | -------------- | -------------- | -------------- |
> | GRF  | Success Rate | **100.0 $\pm$ 0.0** | 98.2  $\pm$ 1.0 | 59.2 $\pm$13.7 | 70.0 $\pm$ 9.8 | 73.5 $\pm$ 8.3 | 88.8 $\pm$ 3.9 |
> | GRF  | Steps Taken  | **25.4 $\pm$ 0.4**  | 34.3 $\pm$ 1.3  | 39.3 $\pm$ 2.4 | 40.4 $\pm$ 1.2 | 41.5 $\pm$ 2.8 | 39.1 $\pm$ 3.1            |
>
> | Task | Metric                    | CommFormer(0.4)    | MAGIC              | HetNet             | CommNet            | I3CNet             | TarMAC             |
> | ---- | ------------------------- | ------------------ | ------------------ | ------------------ | ------------------ | ------------------ | ------------------ |
> | PP   | Average Cumulative Reward | **-0.121 $\pm$ 0.008** | -0.386 $\pm$ 0.024 | -0.232 $\pm$ 0.010 | -0.336 $\pm$ 0.012 | -0.342 $\pm$ 0.015 | -0.563 $\pm$ 0.030 |
> | PP   | Steps Taken               | **4.99 $\pm$ 0.31**    | 10.6 $\pm$ 0.50    | 8.30 $\pm$ 0.25    | 8.97 $\pm$ 0.25    | 9.69 $\pm$ 0.26    | 18.4 $\pm$ 0.46     |
>
> | Task | Metric                    | CommFormer(0.4)    | MAGIC              | HetNet             | CommNet            | I3CNet             | TarMAC             |
> | ---- | ------------------------- | ------------------ | ------------------ | ------------------ | ------------------ | ------------------ | ------------------ |
> | PCP  | Average Cumulative Reward | **-0.197 $\pm$ 0.019** | -0.394 $\pm$ 0.017 | -0.364 $\pm$ 0.017 | -0.394 $\pm$ 0.019 | -0.411 $\pm$ 0.019 | -0.548 $\pm$ 0.031 |
> | PCP  | Steps Taken               | **7.61 $\pm$ 0.66**    | 10.8 $\pm$ 0.45    | 9.98 $\pm$ 0.36    | 11.3 $\pm$ 0.34    | 11.5 $\pm$ 0.37    | 17.0 $\pm$ 0.80    |

---

> ### Author Response · Authors · 2023-11-17
>
> ### Q2
> > In Table 1, the FC is a little bit confusing. Even there are no constrictions on the communication bandwidth, the win rate is still hard to be 100% as it depends how the opponents perform. Of course, 100% is the maximum value for the win rate, but it is a meaningless upper bound. Further, how the value 93.8 is obtained in FC column as the upper bound?
>
> In our study, "FC" refers to CommFormer with a sparsity setting of 1.0. This configuration implies that there are no restrictions on the communication graph, allowing agents to freely communicate with all other agents. Effectively, this represents the upper performance limit of the CommFormer methods. By presenting results under this setting, we aim to demonstrate that with our bi-level learning process, a sparsity of 0.4 can achieve comparable results to a full sparsity of 1.0 in most scenarios.
>
> Given the "FC" framework, the bi-level optimization problem simplifies to the following optimization formulation:
>
> $$
>  \min_{\theta, \phi} ~L_{val}(\phi, \theta)
> $$
>
> ### Q3
> > There have been some works which learns multi-agent cooperative behaviors based on learnable graphs. It would be better to illustrate the differences of the paper compared to them.
>
> Thanks! TWG-Q[12] primarily focuses on exploring diverse spatial-temporal information environments, necessitating the utilization of a temporal weight learning mechanism and weighted GCN to dynamically capture the intensities of cooperations. Conversely, CDC[13] dynamically adjusts the communication graph, taking into account the diffusion process perspective to capture the information flow on the graph. In contrast, our CommFormer approach learns the static communication graph through an optimization perspective, employing attention scores to automatically assign credit to received messages. We will improve the related work in the updated version.
>
> ### Reference
>
> [1] Amanpreet, Singh, et al. "Learning when to communicate at scale in multiagent cooperative and competitive tasks." arXiv preprint arXiv:1812.09755 (2018).
>
> [2] Seraj, Esmaeil, et al. "Learning Efficient Diverse Communication for Cooperative Heterogeneous Teaming." AAMAS 2022.
>
> [3] Xue, Di, et al. "Efficient Multi-Agent Communication via Shapley Message Value." IJCAI 2022.
>
> [4] Das, Abhishek, et al. "Tarmac: Targeted multi-agent communication." ICML 2019.
>
> [5] Wang, Tonghan, et al. "Learning nearly decomposable value functions via communication minimization." arXiv 2019.
>
> [6] Niu, Yaru, et al. "Multi-Agent Graph-Attention Communication and Teaming." AAMAS 2021.
>
> [7] Ryan Kortvelesy and Amanda Prorok. "QGNN: Value Function Factorisation with Graph Neural Networks." arXiv preprint arXiv:2205.13005, 2022.
>
> [8] Sainbayar Sukhbaatar, et al. "Learning multiagent communication with backpropagation." NeurIPS 2016.
>
> [9] Amanpreet Singh, et al. "Learning when to communicate at scale in multiagent cooperative and competitive tasks." arXiv preprint arXiv:1812.09755 (2018).
>
> [10] Yong Liu et al. "Multi-Agent Game Abstraction via Graph Attention Neural Network." AAAI 2022.
>
> [11] Karol Kurach, et al. "Google Research Football: A Novel Reinforcement Learning Environment." AAAI 2020.
>
> [12] Liu, Yuntao, et al. "Temporal Dynamic Weighted Graph Convolution for Multi-agent Reinforcement Learning." Proceedings of the Annual Meeting of the Cognitive Science Society.
>
> [13] Pesce, Emanuele, et al. "Learning multi‑agent coordination through connectivity‑driven communication." Machine Learning Springer 2023.

---

> > ### Author Response · Authors · 2023-11-23
> >
> > Dear Reviewer WiJp,
> >
> > The authors greatly appreciate your time and effort in reviewing this submission, and eagerly await your response. We understand you might be quite busy. However, the discussion deadline is approaching, and we have only a few hours left.
> >
> > We have provided detailed responses to every one of your concerns/questions. Please help us to review our responses once again and kindly let us know whether they fully or partially address your concerns and if our explanations are in the right direction.
> >
> > Best Regards,
> >
> > The authors of Submission 3301

---

### Official Review · Reviewer_Ux29 · 2023-10-31

**Soundness:** 3 good
**Presentation:** 2 fair
**Contribution:** 2 fair
**Rating:** 6
**Confidence:** 5

**Summary:**

The paper introduces CommFormer, a novel approach for optimizing the communication architecture among multiple intelligent agents involved in collaborative tasks. By conceptualizing the architecture as a learnable graph and employing a bi-level optimization process with attention units, CommFormer enables agents to efficiently optimize their communication and adapt more coordinated strategies in a variety of scenarios, as demonstrated in experiments on StarCraft II combat games.

I have several comments and questions that need to be addressed before publication:

- what if the communication graph determined by your approach is not physically feasible, for instance due to environmental constraints such as a far physical distance, etc.? Isn’t a graph communication approach that determines the communication based on physical proximity better in such real-world scenarios? Maybe the best solution is a hybrid approach where environment constraints are considered and baked into the problem for determining the communication graph?

- I find the presented related work section to be weak and relatively old. Many recent SOTA graph-based multi-agent communication learning approaches are never mentioned or discussed, despite their high relevance to the proposed approach. For instance, [1]-[4] below are only a few of such works. Almost all of these works offer a distributed graph-based learned multi-agent communication method that work under POMDPs and are trained under CTDE. There are more of such recent paper. I believe the authors need to perform a more comprehensive search on the recent literature.

[1] Seraj, Esmaeil, et al. "Learning efficient diverse communication for cooperative heterogeneous teaming." Proceedings of the 21st international conference on autonomous agents and multiagent systems. 2022.

[2] Niu, Yaru, Rohan R. Paleja, and Matthew C. Gombolay. "Multi-Agent Graph-Attention Communication and Teaming." AAMAS. 2021.

[3] Bettini, Matteo, Ajay Shankar, and Amanda Prorok. "Heterogeneous Multi-Robot Reinforcement Learning." Proceedings of the 2023 International Conference on Autonomous Agents and Multiagent Systems. 2023.

[4] Meneghetti, Douglas De Rizzo, and Reinaldo Augusto da Costa Bianchi. "Towards heterogeneous multi-agent reinforcement learning with graph neural networks." arXiv preprint arXiv:2009.13161 (2020).

- There are many existing recent, SOTA graph-based multi-agent communication learning approaches, see above [1]-[4] (which are not even mentioned in the paper), that could be a competition for the proposed approach. The selected benchmarks do not necessarily specialize in graph-based distributed communication. The proposed learned communication graph approach should be experimented and evaluated against other graph-based methods.

- All the evaluations are performed in SMAC domains. Is this approach specialized and designed for SMAC? If not, and the solution is in fact generalizable, other domains and different problem settings must be considered. Many of such standard domains can be found in the prior work. Although SMAC domains are interesting game scenarios, the point is to have a comparable baseline performance in standard domains that can also solve other multi-agent coordination and collaboration problems, social interactions, etc.

- Related to the point above, if the presented approach does not apply to other multi-agent problems and scenarios, this should be mentioned and discussed as a limitation. Otherwise, only presenting results in one domain does not suffice.

- The second contribution bullet-point mentions the use of attention units for allocating credit to received messages. Doesn’t TarMAC already do that?

- What are the limitations of the approach? The limitations are never discussed.

At current states I vote weak rejection, since the algorithm seems to be sound and working, however there are some notable weaknesses in literature review and benchmarking (methods and domains) that need to be addressed as much as possible. I’d be happy to increase my score further when authors satisfactorily addressed my comments and questions.

**Strengths:**

See above.

**Weaknesses:**

See above.

**Questions:**

See above.

---

> ### Author Response · Authors · 2023-11-17
>
> ### Q1 & Q2
> >  The proposed learned communication graph approach should be experimented and evaluated against other graph-based methods.
>
> >  Is this approach specialized and designed for SMAC?
>
> Thanks for the suggestion! We have incorporated additional experiments to enhance the generalization of our method. Taking into account the communication domains explored in previous works, we have included the following experiments. It is worth noting that in certain domains, our objective extends beyond maximizing the average success rate or cumulative rewards. We also aim to minimize the average number of steps required to complete an episode, emphasizing the ability to achieve goals in the shortest possible time.
> + Another three maps in the SMAC environment: 1o10b_vs_1r and 1o2r_vs_4r, which pose challenges due to partial observability, and 5z_vs_1ul, where successful outcomes require strong coordination.
> + Predator-Prey (PP) [1]: The goal is for 𝑁 predator agents with limited vision to find a stationary prey and move to its location. The agents in this domain all belong to the same class (i.e., identical state, observation and action spaces).
> + Predator-Capture-Prey (PCP) [2]: We have two classes of predator and capture agents. Agents of the predator class have the goal of finding the prey with limited vision (similar to agents in PP). Agents of the capture class, have the goal of locating the prey and capturing it with an additional capture-prey action in their action-space, while not having any observation inputs (e.g., lack of scanning sensors).
> + Google Research Football (GRF) [11]: We evaluate algorithms in the football academy scenario 3 vs. 2, where we have 3 attackers vs. 1 defender, and 1 goalie. The three offending agents are controlled by the MARL algorithm, and the two defending agents are controlled by a built-in AI. We find that utilizing a 3 vs. 2 scenario challenges the robustness of MARL algorithms to stochasticity and sparse rewards.
>
> We include several state-of-the-art graph-based multi-agent communication learning approaches as additional baselines in our evaluation. These methods encompass QGNN [7], SMS [3], TarMAC [4], NDQ [5], MAGIC [6], HetNet [2], CommNet [8], I3CNet [9], and GA-Comm [10].
>
> The performance results of these baselines are presented below. It is important to note that due to time constraints, we directly obtain the performance results from the respective papers. Our CommFormer consistently demonstrates favorable performance across the evaluated metrics.
>
>
>
> | Task        | Metric       | CommFormer(0.4) | QGNN           | SMS  | TarMAC | NDQ  | MAGIC | QMIX |
> | ----------- | ------------ | --------------- | -------------- | ---- | ------ | ---- | ----- | ---- |
> | 1o2r_vs_4r  | Success Rate | **96.9 $\pm$ 1.5**  | 93.8 $\pm$ 2.6 | 76.4 | 39.1   | 77.1 | 22.3  | 51.1 |
> | 1o10b_vs_1r | Success Rate | 96.9 $\pm$ 3.1  | **98.0 $\pm$ 2.9** | 86.0 | 40.1   | 78.1 | 5.8   | 51.4 |
> | 5z_vs_1ul   | Success Rate | **100.0 $\pm$ 1.4** | 92.2 $\pm$ 1.6 | 59.9 | 44.2   | 48.9 | 0.0   | 82.6 |
>
> | Task | Metric       | CommFormer(0.4) | MAGIC           | CommNet        | I3CNet         | TarMAC         | GA-Comm        |
> | ---- | ------------ | --------------- | --------------- | -------------- | -------------- | -------------- | -------------- |
> | GRF  | Success Rate | **100.0 $\pm$ 0.0** | 98.2  $\pm$ 1.0 | 59.2 $\pm$13.7 | 70.0 $\pm$ 9.8 | 73.5 $\pm$ 8.3 | 88.8 $\pm$ 3.9 |
> | GRF  | Steps Taken  | **25.4 $\pm$ 0.4**  | 34.3 $\pm$ 1.3  | 39.3 $\pm$ 2.4 | 40.4 $\pm$ 1.2 | 41.5 $\pm$ 2.8 | 39.1 $\pm$ 3.1            |
>
> | Task | Metric                    | CommFormer(0.4)    | MAGIC              | HetNet             | CommNet            | I3CNet             | TarMAC             |
> | ---- | ------------------------- | ------------------ | ------------------ | ------------------ | ------------------ | ------------------ | ------------------ |
> | PP   | Average Cumulative Reward | **-0.121 $\pm$ 0.008** | -0.386 $\pm$ 0.024 | -0.232 $\pm$ 0.010 | -0.336 $\pm$ 0.012 | -0.342 $\pm$ 0.015 | -0.563 $\pm$ 0.030 |
> | PP   | Steps Taken               | **4.99 $\pm$ 0.31**    | 10.6 $\pm$ 0.50    | 8.30 $\pm$ 0.25    | 8.97 $\pm$ 0.25    | 9.69 $\pm$ 0.26    | 18.4 $\pm$ 0.46     |
>
> | Task | Metric                    | CommFormer(0.4)    | MAGIC              | HetNet             | CommNet            | I3CNet             | TarMAC             |
> | ---- | ------------------------- | ------------------ | ------------------ | ------------------ | ------------------ | ------------------ | ------------------ |
> | PCP  | Average Cumulative Reward | **-0.197 $\pm$ 0.019** | -0.394 $\pm$ 0.017 | -0.364 $\pm$ 0.017 | -0.394 $\pm$ 0.019 | -0.411 $\pm$ 0.019 | -0.548 $\pm$ 0.031 |
> | PCP  | Steps Taken               | **7.61 $\pm$ 0.66**    | 10.8 $\pm$ 0.45    | 9.98 $\pm$ 0.36    | 11.3 $\pm$ 0.34    | 11.5 $\pm$ 0.37    | 17.0 $\pm$ 0.80    |

---

> > ### Author Response · Authors · 2023-11-17
> >
> > ### Q3
> > > what if the communication graph determined by your approach is not physically feasible, for instance due to environmental constraints such as a far physical distance, etc.? Isn’t a graph communication approach that determines the communication based on physical proximity better in such real-world scenarios? Maybe the best solution is a hybrid approach where environment constraints are considered and baked into the problem for determining the communication graph?
> >
> > Thanks for the valuable comment.
> >
> > A possible application of this study is to create an efficient communication framework tailored for enclosed, finite environments, typical of logistics warehouses. In these settings, agent movement is limited to designated zones, and communication is facilitated through overhead wires, akin to a trolleybus system.
> >
> > In contrast, open environments present unique challenges, primarily due to the potential vast distances between agents, which requires wireless communication and may hinder effective communication. To address this, a straightforward approach could be to add bidirectional edges between agents when they come within close proximity, enabling communication between them [2]. However, a more effective solution may involve a hybrid approach that considers the constraint on the available bandwidth：initially segmenting agents into groups based on proximity, followed by an internal search for an optimal communication graph within each group. If agent distances vary dynamically during testing, this process is repeated as necessary to adjust the communication graph in real-time, ensuring continuous adaptability to changing environmental conditions.
> >
> > ### Q4
> > > The second contribution bullet-point mentions the use of attention units for allocating credit to received messages. Doesn’t TarMAC already do that?
> >
> >
> > In our framework, two attention units are implemented within the encoder and decoder blocks. In the encoder block, the attention unit is tasked with allocating credit to observations received from other agents. This mechanism is somewhat akin to the approach used in TarMAC, which employs targeted, multi-round communication.
> >
> > Conversely, within the decoder block, we introduce a specific constraint. This constraint limits attention computations to interactions between an agent $i$ and its preceding agents $j$, where $j<i$. Such a restriction upholds a sequential update scheme, crucial for the decoder to generate the action sequence in an auto-regressive manner, which ensures a monotonic improvement in performance throughout the training period [12].
> >
> >
> > ### Q5
> > > What are the limitations of the approach? The limitations are never discussed.
> >
> > Thanks for this suggestion! Firstly, our approach is not suitable for deployment in open regions where the physical proximity between agents may exceed the communication range. Additionally, our method may not generalize well to environments that necessitate dynamic communication patterns, such as situations where agents need to interact with different agents at different stages to accomplish tasks. Furthermore, when dealing with a large number of agents, the active edges, determined by the sparsity coefficient, can still impose a physical burden. In such cases, it may be more appropriate to determine a fixed number of edges rather than relying solely on sparsity considerations. These considerations need to be taken into account for further improvements and generalization of our method.
> >
> > ### Q6
> >
> > > There are more of such recent paper. I believe the authors need to perform a more comprehensive search on the recent literature [2, 6, 14, 15].
> >
> > Thanks! These methods primarily employ GNNs to encode pre-defined graphs, allowing for the acquisition of efficient and diverse communication models to facilitate coordination within cooperative and heterogeneous teams. These approaches emphasize the importance of graph feature learning and heterogeneous policy learning, aiming to improve performance in these aspects. In contrast, our CommFormer approach takes a novel approach by simultaneously learning the communication graph and heterogeneous policy from an optimization perspective. By doing so, it seeks to identify and utilize the optimal communication graph for enhanced performance. We will improve the related work in the updated version.

---

> > > ### Author Response · Authors · 2023-11-17
> > >
> > > ### Reference
> > >
> > > [1] Amanpreet, Singh, et al. "Learning when to communicate at scale in multiagent cooperative and competitive tasks." arXiv preprint arXiv:1812.09755 (2018).
> > >
> > > [2] Seraj, Esmaeil, et al. "Learning Efficient Diverse Communication for Cooperative Heterogeneous Teaming." AAMAS 2022.
> > >
> > > [3] Xue, Di, et al. "Efficient Multi-Agent Communication via Shapley Message Value." IJCAI 2022.
> > >
> > > [4] Das, Abhishek, et al. "Tarmac: Targeted multi-agent communication." ICML 2019.
> > >
> > > [5] Wang, Tonghan, et al. "Learning nearly decomposable value functions via communication minimization." arXiv 2019.
> > >
> > > [6] Niu, Yaru, et al. "Multi-Agent Graph-Attention Communication and Teaming." AAMAS 2021.
> > >
> > > [7] Ryan Kortvelesy and Amanda Prorok. "QGNN: Value Function Factorisation with Graph Neural Networks." arXiv preprint arXiv:2205.13005, 2022.
> > >
> > > [8] Sainbayar Sukhbaatar, et al. "Learning multiagent communication with backpropagation." NeurIPS 2016.
> > >
> > > [9] Amanpreet Singh, et al. "Learning when to communicate at scale in multiagent cooperative and competitive tasks." arXiv preprint arXiv:1812.09755 (2018).
> > >
> > > [10] Yong Liu et al. "Multi-Agent Game Abstraction via Graph Attention Neural Network." AAAI 2022.
> > >
> > > [11] Karol Kurach, et al. "Google Research Football: A Novel Reinforcement Learning Environment." AAAI 2020.
> > >
> > > [12] Wen, Muning, et al. “Multi-agent reinforcement learning is a sequence modeling problem.” NeurIPS 2022.
> > >
> > > [14] Bettini, Matteo, Ajay Shankar, and Amanda Prorok. "Heterogeneous Multi-Robot Reinforcement Learning." Proceedings of the 2023 International Conference on Autonomous Agents and Multiagent Systems. 2023.
> > >
> > > [15] Meneghetti, Douglas De Rizzo, and Reinaldo Augusto da Costa Bianchi. "Towards heterogeneous multi-agent reinforcement learning with graph neural networks." arXiv preprint arXiv:2009.13161 (2020).

---

> > > > ### Comment · Reviewer_Ux29 · 2023-11-20
> > > > **Response to Authors**
> > > >
> > > > Thank you to authors for clarifications. While I'm satisfied with most of the responses, I suggest that authors make sure to include the above responses (mainly Q1, Q2, Q3, Q4, and Q5) in the camera-ready version upon acceptance. I believe it is critical to include the new results, domains, and discussions regarding the limitations in open areas as well as differences with the attention mechanism in TarMAC. I also appreciate the additional evaluations and results, which adds to the value of their work.
> > > >
> > > > I raise my score, contingent upon applying the required revisions, as mention above, by the authors.
> > > > Good luck!

---

### Official Review · Reviewer_q3Zy · 2023-10-31

**Soundness:** 3 good
**Presentation:** 3 good
**Contribution:** 2 fair
**Rating:** 8
**Confidence:** 3

**Summary:**

- learning to leverage communication in bandwidth-restricted settings with a learnable adjacency matrix

- continuous relaxation of adjacency matrix to enable differentiable updating of the parameters and adjacency matrix with bootstrapping

**Strengths:**

- Solid formalization of the communication graph problem

- novel contribution**

- impressive experimental results on SMAC compared to SOTA methods

- well written paper, a pleasure to read

** Possible related work: Learning multi-agent coordination through connectivity-driven communication, Pesce and Montana, 2022, springer https://link.springer.com/article/10.1007/s10994-022-06286-6

**Weaknesses:**

- fixed communication network after training. Despite the authors claiming that dynamic adjustments fall outside the scope of the paper, it would be interesting to see performance comparisons.

- task 8m is not in figure 4 (as opposed to what the "Sparsity" paragraph would suggest in 4.3 Ablations).

- "Nevertheless, As task complexity and the number of participating agents increase, a higher
level of sparsity becomes necessary to attain superior performance." this is a very confusing way to say that the matrix needs to be *less* sparse.

**Questions:**

Why do Dynamic adjustments fall outside the scope of the paper? It seems like this is more about considering a simplified problem setting, where the communication graph between training and execution must be similar. Did you run any experiments testing the performance of CommFormer when the nature of the communication graph changes between training and execution?

"where  ̄φ is the target network’s parameter, which is non-differentiable and updated every few epochs" what does this mean?

How does the actual runtime complexity (i.e. walltime or asymptotic) compare between the different methods? S = 0.4 is still quadratic in the number of agents, which can be limiting for large numbers of agents. Rather than having a sparsity proportion, wouldn't it be more relevant to evaluate sparsity as the actual number of non-zero values in the matrix?

Doesn't this method overfit its communication graph to the task? What does a train/test split look like in such a scenario? Do I need to assume with this training method that the communication graph remains the same between training and testing?

Why does additional environment steps seem to solve the sparsity problem in 25m?

In figure 6, any intuition as to what kind of tasks lead Commformer to perform similarly to MAT, and under FC? Since MAT allows unrestricted communication between agents, it's weird that FC seems to massively outperform MAT on some tasks.

---

> ### Author Response · Authors · 2023-11-17
>
> ### Q1
> > Despite the authors claiming that dynamic adjustments fall outside the scope of the paper, it would be interesting to see performance comparisons.
>
>
> Thanks for the suggestion.
>
> The learning process of the communication graph in our study is conducted through bi-level optimization, which necessitates backward propagation of loss. Consequently, during inference, the communication graph remains static and cannot be updated.
>
>
> To adapt our method to a dynamic version, the most straightforward approach is to adjust the communication graph based on the attention scores. We conduct tests on four maps, which involve both homogeneous and heterogeneous agents. The performance results of these experiments are presented below.
>
>
> | Maps | CommFormer(0.4) | Dynamic Version (0.4) |
> | ---- | --------------- | ------------------ |
> | 8m   | 100.0 $\pm$ 0.0 | 96.9 $\pm$ 3.1     |
> | 25m  | 100.0 $\pm$ 0.0 | 71.9 $\pm$ 9.2     |
> | MMM  | 100.0 $\pm$ 0.0 | 100.0 $\pm$ 3.1    |
> | 3s5z | 100.0 $\pm$ 0.0 | 0.0 $\pm$ 0.1      |
>
> As indicated, the dynamic version exhibits a decline in performance across all scenarios. In homogeneous agent environments, the dynamic version demonstrates relatively robust performance. However, in heterogeneous settings, such as in the 3s5z map, it fails to effectively learn communication relationships. This shortfall is likely due to the instability of the training process and a self-boosting phenomenon, where the network is preferentially updated based on relations with initially high attention scores.
>
> We also compare our method with others that specifically investigate dynamic communication adjustments, such as SMS[1], TarMAC[2], and the dynamic message-passing method QGNN[3]. These comparisons are made on three maps: 1o10b_vs_1r and 1o2r_vs_4r, which are challenging due to partial observability, and 5z_vs_1ul, where strong coordination is essential for success. As illustrated in the table below, our method consistently demonstrates exceptional performance.
>
> |             | CommFormer(0.4)     | SMS[1] | TarMAC[2] | QGNN[3]        |
> | ----------- | ------------------- | ------ | --------- | -------------- |
> | 1o10b_vs_1r | 96.9 $\pm$ 3.1      | 86.0   | 40.1      | **98.0 $\pm$ 2.9** |
> | 1o2r_vs_4r  | **96.9 $\pm$ 1.5**  | 76.4   | 39.1      | 93.8 $\pm$ 2.6 |
> | 5z_vs_1ul   | **100.0 $\pm$ 1.4** | 59.9   | 44.2      | 92.2 $\pm$ 1.6 |
>
>
>
> ### Q2
> > Why do Dynamic adjustments fall outside the scope of the paper? It seems like this is more about considering a simplified problem setting, where the communication graph between training and execution must be similar. Did you run any experiments testing the performance of CommFormer when the nature of the communication graph changes between training and execution?
>
> Thank you for this helpful suggestion!
>
> (1) Dynamic adjustments, while ensuring adherence to sparsity requirement at each step, operate under the assumption that all agents require constant communication with one of the other agents. This process typically demands multiple rounds to establish the current communication graph, potentially leading to inefficient bandwidth usage and imposing practical challenges in real-world applications.
>
>
> (2) The primary goal of our research is to identify the most effective communication graph for a given task. During the training phase, we explore the best possible communication graph from a total of  $C(N^2, m)$ potential edge configurations ( $N$ represents the number of agents, and $m$ denotes the number of edges). Upon transition to the execution phase, this communication graph is set and remains unchanged. This fixed communication model is particularly advantageous for practical deployment scenarios.
>
> (3) Our training methodology focuses on identifying the optimal communication graph. To prevent biases towards specific edges (self-boosting phenomenon), we employ the Gumbel-Softmax trick. This approach enables random edge selection during training, ensuring that each potential connection is considered. Consequently, deviating from the determined communication graph during inference can lead to performance degradation. However, due to the comprehensive nature of our training process, the impact on performance might not be drastic. We validate this hypothesis through tests on four maps involving both homogeneous and heterogeneous agents, with the performance outcomes detailed below:
>
> | Maps | CommFormer(0.4) | Graph Changes between T&E|
> | ---- | --------------- | --------------------- |
> | 8m   | 100.0 $\pm$ 0.0 | 93.8 $\pm$ 4.4        |
> | 25m  | 100.0 $\pm$ 0.0 | 95.6 $\pm$ 4.5        |
> | MMM  | 100.0 $\pm$ 0.0 | 93.8 $\pm$ 4.4       |
> | 3s5z | 100.0 $\pm$ 0.0 | 92.6 $\pm$ 4.0        |

---

> > ### Author Response · Authors · 2023-11-17
> >
> > ### Q3
> > > "where ̄φ is the target network’s parameter, which is non-differentiable and updated every few epochs" what does this mean?
> >
> > Sorry for the confusion. In our context, $\phi^-$ refers to the target value function, which is a separate neural network that is a copy of the main value function. It is implemented as an exponential moving average or updated periodically in a "hard" manner. This treatment is similar to how the parameter of the target Q network is handled in DQN (Deep Q-Network) to enhance training stability.
> >
> > The reason the target network is not updated frequently is to address a problem known as "moving target problem" or "target overestimation problem." When the Q-network is updated, the target values used for training are also updated accordingly. If the target network is updated too frequently, the target values can become unstable, leading to slower convergence or even divergence. By keeping the target network fixed for a certain number of iterations, the algorithm can mitigate the instability issue.
> >
> >
> >
> > ### Q4
> > > How does the actual runtime complexity (i.e. walltime or asymptotic) compare between the different methods? S = 0.4 is still quadratic in the number of agents, which can be limiting for large numbers of agents. Rather than having a sparsity proportion, wouldn't it be more relevant to evaluate sparsity as the actual number of non-zero values in the matrix?
> >
> > Thanks for the valuable comment.
> >
> > (1) A feasible approach in our study is to set the total number of communication edges, corresponding to the count of non-zero entries in the matrix. In this context, Equation (8) in the bi-level optimization problem would be reformulated as $|\alpha| \leq m$, without altering the overall training process. The rationale behind considering sparsity primarily relates to the increasing need for communication among agents as their number grows, which in turn could enhance performance. As depicted in Figure 4, a lower sparsity coefficient adversely affects both the learning process and the final outcome. Thus, setting a fixed number $m$ for the edges could lead to reduced performance when the number of agents increases. However, as pointed out by the reviewers, utilizing a fixed number of edges could be more advantageous in scenarios where a large number of agents are involved. This approach takes into account the practical feasibility of maintaining communication between a significant number of agents.
> >
> > (2) Regarding runtime efficiency, action generation, which requires the decoder to produce the action sequence in an auto-regressive manner, is typically the most time-consuming operation in the execution stage. However considering the overall time (communication and execution time), our method is more efficient compared to other communication methods such as TarMAC and SMS, which necessitate multiple rounds of information exchange, or QGNN, which requires multiple rounds of information transfer through a GNN model. Our method, which involves a one-time transfer of local observation and action sequences, is both straightforward and time-saving. Determining the theoretical complexity of each method is challenging owing to the intricate engineering design involved. Due to constraints such as limited time and the absence of open-source code, we substantiate our assertions using empirical runtime data. For instance, in the 5z_vs_1ul map scenario, our method accomplish training within 8 hours, whereas QGNN required nearly 2 days to accomplish training process.
> >
> > ### Q5
> > > Doesn't this method overfit its communication graph to the task? What does a train/test split look like in such a scenario? Do I need to assume with this training method that the communication graph remains the same between training and testing?
> >
> >
> > Our study aims to identify the optimal communication graph for specific tasks. As illustrated in Figure 1, manually determining the communication graph requires meticulous design, as an inadequately conceived graph could lead to suboptimal performance. Therefore, we approach this challenge by conceptualizing the task as finding the most effective communication graph, while simultaneously allowing for normal updates of the architectural parameters in an end-to-end manner.
> >
> > All our experiments are conducted in an online setting, meaning that training and testing for each scenario occur within the same map. However, there is a distinct difference between the training and inference phases. During training, every agent is permitted to communicate with all others, facilitating the discovery of the most suitable communication graph. In contrast, the inference phase restricts each agent to communicate only with a limited set of agents, as determined by the established communication graph. This process adheres to the principle of centralized training and decentralized execution (CTDE).

---

> > > ### Author Response · Authors · 2023-11-17
> > >
> > > ### Q6
> > > > Why does additional environment steps seem to solve the sparsity problem in 25m?
> > >
> > > Based on the observations from Figure 4, when the sparsity coefficient $S$ is low (e.g., 0.1), the process of determining the communication graph may require more steps. This is because each edge becomes crucial due to the limited total number of edges. However, once the algorithm identifies the truly significant edges, it can achieve improved performance in subsequent steps. Therefore, it may take more iterations to identify and prioritize the important edges when the sparsity coefficient is low.
> > >
> > > ### Q7
> > > > In figure 6, any intuition as to what kind of tasks lead Commformer to perform similarly to MAT, and under FC? Since MAT allows unrestricted communication between agents, it's weird that FC seems to massively outperform MAT on some tasks.
> > >
> > > In our analysis, we compare our method with MAT, specifically focusing on its fully decentralized actor variant for each individual agent, as outlined in the MAT-dec of their original paper. This comparison ensures a fair assessment of rest approaches. Additionally, in our study, "FC" refers to CommFormer with a sparsity setting of 1.0. This configuration implies that there are no restrictions on the communication graph, allowing agents to freely communicate with all other agents. Effectively, this represents the upper performance limit of the CommFormer methods. By presenting results under this setting, we aim to demonstrate that with our bi-level learning process, a sparsity of 0.4 can achieve comparable results to a full sparsity of 1.0 in most scenarios.
> > >
> > > Given the "FC" framework, the bi-level optimization problem simplifies to the following optimization formulation:
> > >
> > > $$
> > >  \min_{\theta, \phi} ~L_{val}(\phi, \theta)
> > > $$
> > >
> > >
> > > The performance of CommFormer varies depending on the nature of the task. In scenarios where each agent primarily relies on its own capabilities and communication plays a less critical role, CommFormer's performance is comparable to that of MAT. However, in situations where inter-agent communication is crucial for success, MAT tends to struggle in achieving optimal performance, whereas CommFormer is designed to excel in these contexts by effectively leveraging communication strategies.
> > >
> > >
> > > ### Q8
> > > > related work: Learning multi-agent coordination through connectivity-driven communication
> > >
> > >
> > > > task 8m is not in figure 4 (as opposed to what the "Sparsity" paragraph would suggest in 4.3 Ablations).
> > >
> > > > "Nevertheless, As task complexity and the number of participating agents increase, a higher level of sparsity becomes necessary to attain superior performance." this is a very confusing way to say that the matrix needs to be *less* sparse.
> > >
> > > Thanks for pointing these out. CDC, similar to our CommFormer approach, represents agents as nodes and employs graph-dependent attention mechanisms to govern the weighting of incoming messages from agents. However, CDC dynamically modifies the communication graph based on a diffusion process perspective, whereas our CommFormer learns the static communication graph from an optimization standpoint. We will add the missing figure, improve the statement, and add the missing citations in the updated version.
> > >
> > > ### Reference
> > >
> > > [1] Xue, Di, et al. “Efficient Multi-Agent Communication via Shapley Message Value.” IJCAI 2022.
> > >
> > > [2] Das, Abhishek, et al. “Tarmac: Targeted multi-agent communication.” ICML 2019.
> > >
> > > [3] Ryan Kortvelesy and Amanda Prorok. “QGNN: Value Function Factorisation with Graph Neural Networks.” arXiv preprint arXiv:2205.13005, 2022.

---

### Official Review · Reviewer_9cgB · 2023-11-01

**Soundness:** 4 excellent
**Presentation:** 4 excellent
**Contribution:** 4 excellent
**Rating:** 8
**Confidence:** 5

**Summary:**

This paper proposes a method for learning optimal communication graphs in multi-agent systems using attention. Unlike previous methods that use a predefined graph communication structure with unlimited comms bandwidth, CommFormer learns to create directed communication links such that some level of graph sparsity $S$ is maintained. To do this, they formulate a constrained optimization problem to learn a value encoder and action decoder with an upper bound on the norm of the graph adjacency matrix. In reality, they create a bi-level optimization that steps the encoder/decoder optimizers to find approximate optima then update the adjacency matrix. They perform experiments on StarCraftII with various value-based and policy gradient-based baselines and demonstrate that ComFormer can outperform the baselines in SMAC tasks ranging from Easy to Super Hard.

**Strengths:**

This is an interesting and well written paper. To my knowledge, the learned graph for graph communication using transformers is a novel idea with clear applications to the real world. The architecture is simple/clear and the motivation for the necessity of this solution is motivated very well in Figure 1.

1. The CommFormer method significantly outperforms most of the baselines on most of the tasks (with the exception of some super hard SMAC tasks)
2. Performs ablative studies to demonstrate the importance of the sparsity claimed in the paper.
3. Adaptable to various actor-critic methods, not just PPO

**Weaknesses:**

There are some concerns I have about the problem formulation. It is assumed in many MARL tasks that communication at test time is limited, as per the CTDE paradigm. However, my understanding is that at each time step, the CommFormer can choose to create/destroy communication links between any arbitrary agents as long as a sparsity measure is met. While this is not unreasonable, it is a very large assumption to make while claiming the CTDE paradigm. Further, in seciton 3.2, the authors state that they restrict communication of agent $i$ to only agents $j$ where $j< i$; this assumes that there is some implicit (or explicit) ordering of the agents that we are assuming. Again, I don't think this is unreasonable as many MARL algorithms use one-hot agent id encoding, it imposes additional nuances that are important to the functioning of the algorithm.

Finally, the authors do not compare to a recent graph-based MARL baseline called QGNN[1]

[1] Ryan Kortvelesy and Amanda Prorok.  QGNN: Value Function Factorisation with Graph Neural Networks

**Questions:**

1. Can the authors compare their method to QGNN
2. How is the ordering of agents decided when inputting to the transformer and is there positional encoding?
3. I understand that graph sparsity if a necessary assumption to manage the bandwidth of any given agent. However, can the authors discuss or demonstrate what would happen if more realistic assumptions on graph communication were made, such as only communicating with agents within some specified communication range?

---

> ### Author Response · Authors · 2023-11-17
>
> ### Q1
> > How is the ordering of agents decided when inputting to the transformer and is there positional encoding?
>
> Sorry for the misleading. It is important to clarify that we do not manually determine the order of the agents, nor is there any positional encoding specifically assigned to the agents. Instead, we calculate the attention between agent $i$ and its preceding agents $j$ where $j<i$ in order to ensure that the decoder generates the action sequence in an auto-regressive manner. This approach guarantees a consistent improvement in performance during training [1].
>
> In practice, it is possible to change the order of agents at each update iteration. However, we have chosen to maintain the current representation (using $1,2, \dots n$ other than $i_1, i_2, \dots i_n$ ) primarily for the purpose of preserving the order of the adjacency matrix. Altering the order of agents would require corresponding adjustments to the adjacency matrix $\alpha$.
>
>
> ### Q2
> > Compare to a recent graph-based MARL baseline called QGNN.
>
> Thank you for your comment.
>
> QGNN primarily leverages graph pooling to facilitate the process of value factorization within a system of agents that exhibit variable sizes. In this system, the communication graph is predetermined by a specific protocol, such as known interactions, and is employed to represent the interdependencies present within a multi-layer message-passing GNN architecture. On the other hand, our CommFormer algorithm simultaneously determines the communication graph and autonomously assigns credit to received messages through a single round of communication. The requested results have been presented in the table provided below. To consider the time constraints, we have chosen StarcraftII as the environment for testing these methods. The evaluation is conducted on three maps: 1o10b_vs_1r and 1o2r_vs_4r, which pose challenges due to partial observability, and 5z_vs_1ul, where successful outcomes require strong coordination.
>
> To ensure a comprehensive comparison, we have also included the performance of other communication methods, such as SMS[2], TarMAC[3], NDQ[4] and MAGIC[5]. It is worth noting that in the QGNN approach, the graph is fully-connected, following their official code settings. This means that each agent can communicate information with all other agents. CommFormer consistently demonstrates strong performance across these environments.
>
>
> |             | CommFormer(0.4)      | QGNN                | SMS[2] | TarMAC[3] | NDQ[4] | MAGIC[5] | QMIX |
> | ----------- | -------------------- | ------------------- | ------ | --------- | ------ | -------- | ---- |
> | 1o10b_vs_1r | 96.9 $\pm$ 3.1       | **98.0 $\pm$ 2.9** | 86.0   | 40.1      | 78.1   | 5.8      | 51.4 |
> | 1o2r_vs_4r  | **96.9 $\pm$ 1.5**  | 93.8 $\pm$ 2.6      | 76.4   | 39.1      | 77.1   | 22.3     | 51.1 |
> | 5z_vs_1ul   | **100.0 $\pm$ 1.4** | 92.2 $\pm$ 1.6                    | 59.9   | 44.2      | 48.9   | 0.0      | 82.6 |
>
>
> ### Q3
> >  Discuss or demonstrate what would happen if more realistic assumptions on graph communication were made, such as only communicating with agents within some specified communication range?
>
> A possible application of this study is to create an efficient communication framework tailored for enclosed, finite environments, typical of logistics warehouses. In these settings, agent movement is limited to designated zones, and communication is facilitated through overhead wires, akin to a trolleybus system.
>
> Our objective encompasses determining the optimal communication graph, while concurrently ensuring normal updates of architectural parameters. Upon completion of the training phase, the communication graph becomes static, forming the basis for subsequent inferences.
>
> In contrast, open environments present unique challenges, primarily due to the potential vast distances between agents, which requires wireless communication and may hinder effective communication. To address this, a straightforward approach could be to add bidirectional edges between agents when they come within close proximity, enabling communication between them [6]. However, a more effective solution may involve a hybrid approach that considers the constraint on the available bandwidth：initially segmenting agents into groups based on proximity, followed by an internal search for an optimal communication graph within each group. If agent distances vary dynamically during testing, this process is repeated as necessary to adjust the communication graph in real-time, ensuring continuous adaptability to changing environmental conditions.

---

> > ### Author Response · Authors · 2023-11-17
> >
> > ### Q4
> > > At each time step, the CommFormer can choose to create/destroy communication links between any arbitrary agents as long as a sparsity measure is met. While this is not unreasonable, it is a very large assumption to make while claiming the CTDE paradigm.
> >
> >
> > Thanks for pointing these out. In addressing the practical deployment of our CommFormer model in real-world conditions, we consider two approaches based on the available resources and constraints:
> >
> > (1) Simulation-Based Communication Graph Optimization: When a simulation environment closely mirroring the real-world deployment is available, we suggest leveraging this simulation to optimize the communication graph. This method efficiently circumvents the complexities associated with the direct creation and destruction of communication links in a live setting, which only needs to maintain a global adjacency matrix used as the mask. By utilizing a simulated environment, we can iteratively refine and determine the most effective communication graph without the immediate concerns of real-world deployment challenges.
> >
> > (2) Physical Interaction-Based Approach with a Global Bus System (with overhead wires, wireless situation is similar): In scenarios where direct physical interaction is necessary, and a simulation environment is not available, we propose a feasible solution that involves connecting each agent through a global bus system. This setup would allow the dynamic establishment and disbandment of communication links, analogous to how TCP (Transmission Control Protocol) connections operate in network communications. Under this framework, each agent can dynamically establish or terminate connections based on the optimized communication strategy derived from our model.
> >
> > ### Reference
> > [1] Wen, Muning, et al. "Multi-agent reinforcement learning is a sequence modeling problem." NeurIPS 2022.
> >
> > [2] Xue, Di, et al. "Efficient Multi-Agent Communication via Shapley Message Value." IJCAI 2022.
> >
> > [3] Das, Abhishek, et al. "Tarmac: Targeted multi-agent communication." ICML 2019.
> >
> > [4] Wang, Tonghan, et al. "Learning nearly decomposable value functions via communication minimization." arXiv 2019.
> >
> > [5] Niu, Yaru, et al. "Multi-Agent Graph-Attention Communication and Teaming." AAMAS 2021.
> >
> > [6] Seraj, Esmaeil, et al. “Learning Efficient Diverse Communication for Cooperative Heterogeneous Teaming.” AAMAS 2022.

---

### Author Response · Authors · 2023-11-17

### Summary
We thank reviewers for their valuable feedback, and appreciate the great efforts made by all reviewers, ACs, SACs and PCs.

We are invigorated by the **positive evaluation** from all reviewers. Specifically, they find the method **novel and well-motivated**(all), the experimental results being **state-of-the-art**(9cgB, q3Zy), the writing **effectively illustrated**(q3Zy).

In response to the comments and suggestions, we have provided a detailed respective rebuttal for each reviewer, and here we summarize major points for convenience.
+ We have conducted a more in-depth and comprehensive analysis of experimental design and results, including more graph-based communication methods and more communication environments.(9cgB, Ux29,  WiJp)
+ We have extended ablation studies to investigate the effect of dynamic version and graph communication changing version. (q3Zy)

All these will be merged into the article.

---

### Meta-Review · Area_Chair_ouSE · 2023-12-10

**Metareview:**

This paper proposes CommFormer approach for learning communication graphs in multi-agent systems. There raised some concerns regarding comparisons to recent graph-based MARL methods, generalization beyond StarCraft II environments, feasibility for real-world deployment with dynamic communication, differences from prior attention-based communication approaches like TarMAC. In response, the authors have added experiments on several new environments including Predator-Prey, Predator-Capture-Prey, and Google Football, comparing CommFormer to a range of graph-based MARL baselines like QGNN, HetNet, and MAGIC. They have also discussed limitations and provided suggestions for dynamic communication in real-world settings, such as adding edges based on proximity. And they clarified differences in the attention mechanisms between CommFormer and prior work like TarMAC. Overall, the authors have addressed the major concerns through the discussion, demonstrating the competitiveness of CommFormer against graph-based MARL methods and its potential for broader applications.

**Justification For Why Not Higher Score:**

The reviewers seem mostly satisfied with the responses, pending inclusion of the additional results and discussions.

**Justification For Why Not Lower Score:**

The reviewers all acknowledge the contribution of this paper.

---

### Decision · Program_Chairs · 2024-01-16

Accept (poster)